# Usage-Aware Sentiment Representations in Large Language Models

## Abstract

Large language models (LLMs) can encode high-level concepts as linear directions in their representation space, and sentiment has been studied in this framework. However, probe-derived sentiment directions often vary substantially across datasets, thereby compromising reliability for downstream applications. Prior work addresses this issue with distributional methods such as Gaussian subspaces, which improve reliability but trade off direct interpretability of linguistic meaning. In this paper, we propose a usage-aware sentiment representation framework that grounds sentiment variability in linguistic usage factors such as tone, topic, context, and genre, which are drawn from linguistic research. Our framework operates at two complementary levels of analysis: At the axis level, we construct sentiment directions from both pooled and usage-specific data to investigate the role of usage in shaping sentiment representations. At the neuron level, we provide a finer view by distinguishing usage-invariant neurons that consistently encode sentiment from usage-sensitive neurons whose contributions vary across usages. Experiments indicate that usage-aware sentiment representation enhances reliability, improving both classification accuracy and controllability of sentiment steering. Finally, preliminary experiments with audio LLMs suggest that our framework generalizes beyond text, pointing toward cross-modal applicability.

## 1 Introduction

Large language models (LLMs) have rapidly advanced capabilities in language understanding and generation (Radford et al., 2019; Brown et al., 2020). They are increasingly applied in domains where sentiment is central, such as mental health support and psychotherapy assistance (Yang et al., 2023; Gabriel et al., 2024; Hu et al., 2025). In these settings, both reliability and interpretability are crucial. Despite their widespread use, how LLMs internally encode sentiment remains underexplored. A promising direction is probing methods, grounded in the linear representation hypothesis: high-level semantic attributes can be captured as linear directions in representation space (Mikolov et al., 2013; Park et al., 2024; Jiang et al., 2024). This approach has been validated across diverse abstract concepts, including political stance (Kim et al., 2025), refusal (Arditi et al., 2024), and sentiment polarity (Tigges et al., 2024; Di Palma et al., 2025).

However, sentiment directions obtained through probing are often unstable: the axes derived from different datasets can diverge substantially (Figure 1), undermining reliability for downstream applications. Prior work addresses this challenge by expanding single directions into Gaussian subspaces, thereby modeling variability through multiple latent axes (Zhao et al., 2025). Although such distributional approaches improve reliability, they sacrifice interpretability by abstracting away from explicit linguistic meaning. This highlights the need for a framework that captures sentiment variability in a way that is both reliable and linguistically interpretable.

We argue that a natural explanation for variability lies in linguistic usage. Sentiment is rarely expressed in isolation: its manifestation depends on tone, topic, context, and genre. These usage factors have been extensively recognized as shaping sentiment expression in human communication (Ousidhoum et al., 2019; Blitzer et al., 2007; Joshi et al., 2015; Barnes et al., 2017). Following Wittgenstein's dictum that "meaning is use" (Wittgenstein, 1953), we ground sentiment probing in a small set of usage factors drawn from linguistic research. This alignment connects sentiment rep-

resentations to the way sentiment is expressed in language, improving interpretability and strengthening reliability for downstream applications.

Accordingly, we propose a usage-aware sentiment representation framework that explicitly incorporates linguistic usage factors. This framework aims to make sentiment probing both more interpretable and more reliable. Our study proceeds in three stages. First, we prompt LLMs to generate sentiment data covering four usage factors and apply linear probing (Alain & Bengio, 2016; Ousidhoum et al., 2021; Belinkov, 2022) to extract sentiment directions. The main axis is trained on data pooled across all usages, while usage-specific axes are trained on each dimension separately. We further examine whether combining them yields more reliable and complementary representations. Second, we analyze usage-invariant (which encode sentiment consistently across usages) and usage-sensitive (whose contribution to sentiment varies with usage factors) neurons to gain finer-grained insights into how sentiment is shaped by usage factors. Finally, we assess the reliability of the learned representations through two complementary tasks: cross-domain sentiment classification, which tests transferability across datasets, and sentiment steering, which evaluates whether the axes can reliably control model outputs. In addition, we extend our study to audio LLMs (Du et al., 2024), suggesting that usage-aware sentiment axes generalize beyond text to other modalities. Our **contributions** are as follows:

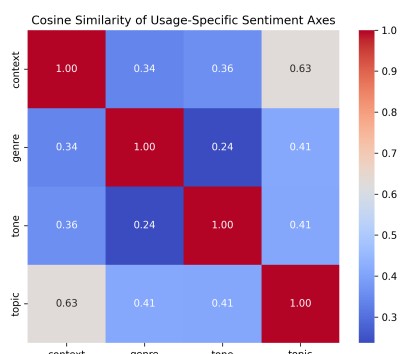

**Figure 1:** Sentiment directions derived from different datasets exhibit low similarity.

- We propose a usage-aware approach that grounds sentiment variability in explicit linguistic factors (tone, topic, context, and genre), providing a linguistically motivated perspective on sentiment representations.
- We operationalize this framework through two strategies: augmenting the main axis with usage-specific axes, and constructing axes from usage-invariant and usage-sensitive neurons, which jointly enhance interpretability and reliability.
- Extensive evaluations on cross-domain classification and controllable sentiment steering show that usage-aware axes yield more reliable and controllable sentiment representations.
- We provide initial evidence that the usage-aware approach generalizes beyond text to audio LLMs, pointing toward cross-modal applicability.

## 2 PRELIMINARIES

**Representation Space of LLMs**  Given tokens $(x_{1:n}) \in \mathcal{V}^n$, we denote the hidden state of token $j$ at layer $\ell$ as $h_j^{(\ell)} \in \mathbb{R}^d$, and collect them as $H^{(\ell)} = [h_1^{(\ell)}, \ldots, h_n^{(\ell)}]^\top \in \mathbb{R}^{n \times d}$. In a decoder-only architecture, hidden states are updated layer by layer through self-attention and feed-forward sublayers with residual connections:

$$\tilde{h}_j^{(\ell)} = h_j^{(\ell)} + \text{SelfAttn}^{(\ell)}(H^{(\ell)})_j, \qquad h_j^{(\ell+1)} = \tilde{h}_j^{(\ell)} + \text{MLP}^{(\ell)}(\tilde{h}_j^{(\ell)}) \qquad (1)$$

To obtain an utterance-level representation at layer $\ell$, we apply mean pooling, since sentiment cues are distributed across tokens and contexts rather than confined to explicit sentiment words (Tigges et al., 2024): $u^{(\ell)} = \frac{1}{n} \sum_{j=1}^n h_j^{(\ell)}$

**Linear Probing**  To analyze what kind of information is captured in model representations across layers, we adopt linear probing (Alain & Bengio, 2016; Belinkov, 2022). Linear probes are lightweight classifiers trained on frozen model activations to test whether a target property can be linearly predicted. This technique has been widely used (Li et al., 2023; Marks & Tegmark, 2024) as it offers a simple diagnostic: if a concept can be decoded with a linear model, it suggests that the information is explicitly encoded in the representation space rather than buried in higher-order nonlinear interactions. Given the utterance-level representation $u^{(\ell)}$ defined above, we train a logistic regression classifier to predict whether an input instance belongs to a target category, such as positive or negative sentiment. Here, $i \in \{1, \ldots, N\}$ indexes training instances (utterances), each

represented by $u_i^{(\ell)}$ at layer $\ell$. The classifier then predicts $\hat{y}_i = \sigma(w^\top u_i^{(\ell)})$ where $w$ is a learnable weight vector and $\sigma(\cdot)$ is the sigmoid function that maps the score into a probability. The probe is trained using standard cross-entropy loss with $\ell_2$-regularization:

$$\mathcal{L}(w) = -\frac{1}{N} \sum_{i=1}^{N} \left( y_i \log \hat{y}_i + (1 - y_i) \log(1 - \hat{y}_i) \right) + \frac{\lambda}{2} \|w\|_2^2 \qquad (2)$$

After training, the normalized weight vector $\hat{w} = w/\|w\|$ can be interpreted as a concept axis in the representation space. In our case, this axis corresponds to sentiment polarity: it points from negative to positive sentiment.

**Steering** Given a sentiment axis $\hat{w}$ obtained via linear probing, we investigate its functional role in the model by applying activation steering. At inference time, this is done by adding the axis vector to hidden activations (Meng et al., 2022). Formally, for the hidden state $h_n^\ell$ of the last token at layer $\ell$, the steered representation is $h_{n,\text{steered}}^\ell = h_n^\ell + \alpha \text{RMS}(h_n^\ell)\, \hat{w}$ where $\alpha \in \mathbb{R}$ controls the intervention strength, $\text{RMS}(h_n^\ell)$ denotes the root-mean-square of the hidden vector, which scales the intervention relative to the activation magnitude, and $\hat{w}$ denotes the chosen sentiment axis (main, usage-specific, or their combination). By varying $\alpha$, we test whether sentiment axes correspond to consistent behavioral shifts, to demonstrate their reliability and interpretability.

## 3 Sentiment-Guided Axis- and Neuron-level Analysis

We introduce a two-level methodology for usage-aware sentiment representation in LLMs that captures both global and fine-grained effects. At the axis level, we compare sentiment directions derived from pooled and usage-specific data to examine how linguistic usage factors shape representational stability and variation. At the neuron level, we further decompose representations into usage-invariant neurons, which encode sentiment consistently across factors, and usage-sensitive neurons, whose contributions vary with usage.

**Axis-level Analysis** We construct a dataset annotated with four usage factors: genre, tone, context, and topic. For each usage $u \in \{\text{genre}, \text{tone}, \text{context}, \text{topic}\}$, we collect a subset $\mathcal{D}u$ containing both positive and negative sentiment instances. The complete dataset is then defined as the union of all subsets: $\mathcal{D} = \bigcup u \mathcal{D}_u$.

On this basis, we train linear probes layer by layer to obtain sentiment axes (as described in Section 2). At each layer $\ell$, the main axis is trained on the full dataset $\mathcal{D}$, yielding a normalized weight vector $\hat{w}_{\text{main}}^\ell = w^\ell(\mathcal{D})/\|w^\ell(\mathcal{D})\|$. In parallel, for each usage $u$, we train a separate probe on $\mathcal{D}_u$ to derive a usage-specific axis $\hat{w}_u^\ell = w^\ell(\mathcal{D}_u)/\|w^\ell(\mathcal{D}_u)\|$. This procedure yields, for each layer, one main axis trained on the full dataset and four usage-specific axes trained on genre, tone, context, and topic. For an LLM with $N$ layers, this results in a total of $5N$ axes. Together, this set of axes enables us to test the influence of usage factors on sentiment encoding and to capture complementary perspectives on sentiment representation across conditions.

**Neuron-level Analysis** To obtain a finer-grained view of usage effects on sentiment, we complement the axis-level analysis with a neuron-level perspective. Each layer of the model contains $n$ neurons, and we compute the polarity of every neuron under different usage conditions. For each neuron $j$ and usage $u$, we compute its mean activation on positive and negative subsets:

$$\mu_{j,+}^{(u)} = \frac{1}{|D_+^{(u)}|} \sum_{x \in D_+^{(u)}} a_j(x), \quad \mu_{j,-}^{(u)} = \frac{1}{|D_-^{(u)}|} \sum_{x \in D_-^{(u)}} a_j(x) \qquad (3)$$

where $a_j(x)$ is the activation of neuron $j$ for input $x$, and $D_+^{(u)}$ and $D_-^{(u)}$ are the positive and negative subsets of usage $u$. The difference $\mu_{j,+}^{(u)} - \mu_{j,-}^{(u)}$ reflects the polarity preference of neuron $j$ under usage $u$, and we define its sign as the polarity label $s_j^{(u)}$. Using these polarity labels, We classify neurons into two categories: usage-invariant (stable) if their polarity remains consistent across all usages, and usage-sensitive (flipped) if polarity differs across usages (e.g., positive under some usages but negative under others). In practice, we consider a neuron flipped only when polarity disagreement is balanced across usages, requiring at least two positive and two negative polarities

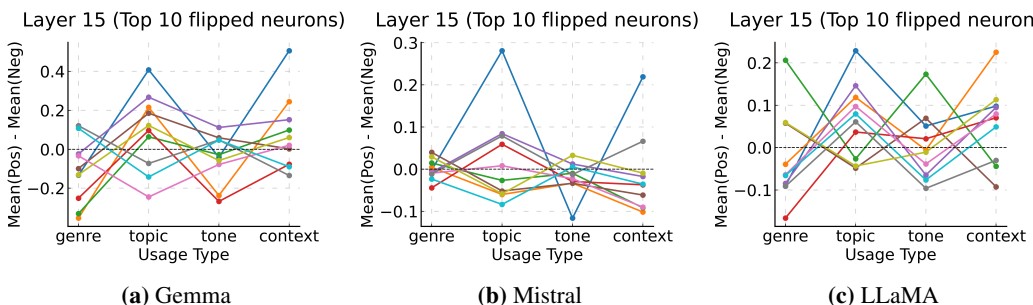

**Figure 2:** Polarity variation of the top 10 flipped neurons at Layer 15 across usage factors in three LLMs, illustrating instability across usages.

(e.g., $|u \mid s_j^{(u)} = +1| = |u \mid s_j^{(u)} = -1| = 2$ for the four usage factors). This stricter criterion filters out incidental polarity shifts, isolating neurons that robustly capture systematic usage sensitivity.

To validate this phenomenon, we visualize the top 10 flipped neurons at Layer 15, selected based on the largest absolute polarity variation across usages. Figure 2 shows clear polarity changes, confirming the presence of usage-sensitive neurons. Finally, to connect back to axis-level analysis, we construct new sentiment axes by masking or retraining on subsets of neurons—only stable, only flipped, or both. This design enables us to directly compare how usage-invariant and usage-sensitive neurons contribute to sentiment representation under both masked and retrained settings.

## 4 SENTIMENT CLASSIFICATION WITH USAGE-AWARE AXES

In this section, we evaluate usage-aware sentiment axes at two levels: axis-level probing, contrasting pooled with usage-specific directions, and neuron-level probing, distinguishing stable from flipped neurons. This two-level analysis provides both a global and fine-grained view of how usage factors shape sentiment encoding, and together they show that usage-aware axes improve the reliability of sentiment classification.

**Data** For training, we construct a synthetic usage-annotated dataset using ChatGPT-4o (OpenAI, 2024). Specifically, we generate four types of usage data: genre, tone, context, and topic. For each usage type, we produce 2,000 samples, resulting in a total of 8,000 training instances. To ensure quality, we manually inspected a random subset, confirming diversity and correctness of sentiment labels. The detailed generation procedure is provided in Appendix A. For evaluation, we uses eight out-of-domain sentiment datasets. These include four standard benchmarks (IMDb (Maas et al., 2011), SST5 (binary version) (Socher et al., 2013), Twitter (Go et al., 2009), and DailyDialog (Li et al., 2017)) and four domain-specific subsets of GoEmotions (Demszky et al., 2020): AnimalsBeingBros (Animals), Confession (Conf), Cringe (Cri), and OkCupid (OkC). Further statistics of these evaluation datasets are provided in Appendix B. We choose these subsets to reflect diverse domains and sentiment expressions: AnimalsBeingBros conveys lighthearted content, Confession captures private emotional disclosure, Cringe illustrates socially awkward scenarios, and OkCupid represents relationship-oriented dialogue.

**Evaluation Setup** For a given input sentence, we first compute its mean-pooled representation $h^\ell$ at layer $\ell$. The sentiment prediction is then obtained by projecting $h^\ell$ onto the corresponding axis $\hat{w}^\ell$. Classification accuracy is reported as the proportion of correct predictions and serves as a proxy for how well the learned axis captures sentiment polarity.

### 4.1 AXIS-LEVEL PROBING

**Training Probes** Following Section 3, we train probes at each layer $\ell$ to derive one main axis from the full dataset $\mathcal{D}$ and four usage-specific axes from subsets $\mathcal{D}_u$ ($u \in \{\text{genre}, \text{tone}, \text{context}, \text{topic}\}$). In all experiments, we set the $\ell_2$ regularization strength to $\lambda = 1$ (Eq. 2). We train probes independently at every transformer layer, yielding one main and four usage-specific axes per layer. For classification, predictions are obtained by projecting representations onto the corresponding axis.

**Table 1:** Out-of-domain sentiment classification accuracy (%) with Main axis and Main+Sub-axis for three LLMs. $\Delta$ = improvement of Main+Sub-axis over Main. Conf = Confession; Cri = Cringe; OkC = OkCupid.

| Method | IMDb | SST5 | Twitter | Dialogue | Animals | Conf | Cri | OkC |
|---|---|---|---|---|---|---|---|---|
| LLaMA Main | 78.15 | 83.98 | 87.95 | 87.40 | 82.86 | 71.76 | 62.23 | 72.33 |
| LLaMA Main+Sub | 81.28 | 81.98 | 90.29 | 88.79 | 85.14 | 77.65 | 73.94 | 76.73 |
| $\Delta$ | **+3.13** | -2.00 | **+2.34** | **+1.39** | **+2.28** | **+5.89** | **+11.71** | **+4.40** |
| Gemma Main | 78.38 | 79.58 | 83.06 | 79.11 | 74.86 | 65.88 | 61.70 | 63.52 |
| Gemma Main+Sub | 83.08 | 80.63 | 87.84 | 85.18 | 82.86 | 74.71 | 68.62 | 75.47 |
| $\Delta$ | **+4.70** | **+1.05** | **+4.78** | **+6.07** | **+8.00** | **+8.83** | **+6.92** | **+11.95** |
| Mistral Main | 74.22 | 77.02 | 88.66 | 86.79 | 81.14 | 72.94 | 68.09 | 69.81 |
| Mistral Main+Sub | 78.57 | 81.69 | 90.72 | 87.56 | 82.29 | 77.65 | 72.34 | 76.73 |
| $\Delta$ | **+4.35** | **+4.67** | **+2.06** | **+0.77** | **+1.15** | **+4.71** | **+4.25** | **+6.92** |
| Average Main | 76.92 | 80.19 | 86.56 | 84.43 | 79.62 | 70.19 | 64.01 | 68.55 |
| Average M+Sub | 80.98 | 81.43 | 89.62 | 87.18 | 83.43 | 76.67 | 71.63 | 76.31 |
| Average $\Delta$ | **+4.06** | **+1.24** | **+3.06** | **+2.75** | **+3.81** | **+6.48** | **+7.62** | **+7.76** |

**Table 2:** Best accuracy (%) and Main+Sub combinations for three LLMs.

| Dataset | LLaMA | Gemma | Mistral |
|---|---|---|---|
| SST5 | 83.98 / main | 80.63 / main+genre+topic | 81.69 / main+topic |
| IMDb | 81.28 / main+context | 83.08 / main+context | 78.57 / main+context |
| Twitter | 90.29 / main+genre+topic | 87.84 / main+topic | 90.72 / main+topic |
| Dialogue | 88.79 / main+genre+topic | 85.18 / main+genre | 87.56 / main+genre+topic |
| Animals | 85.14 / main+genre | 82.86 / main+topic | 82.29 / main+genre+topic |
| Conf | 77.65 / main+genre | 74.71 / main+genre+topic | 77.65 / main+genre+topic+tone |
| Cringe | 73.94 / main+genre | 68.62 / main+genre | 72.34 / main+genre |
| OkCupid | 76.73 / main+genre+topic | 75.47 / main+genre+topic | 76.73 / main+genre |

**Results** Table 1 reports out-of-domain sentiment classification across eight datasets for three LLMs: LLaMA-3-8B-Instruct (Dubey et al., 2024), Gemma-7B-Instruct (Team et al., 2024), and Mistral-7B-Instruct (Jiang et al., 2023). For brevity, we abbreviate them as LLaMA, Gemma, and Mistral. For each model, we report the best result over all layers. Here, Main refers to axes trained on the full dataset, Sub to axes trained on individual usage subsets, and Main+Sub to their combinations (see Table 2 for details of the combination procedure). See Appendix C for model specifications and Appendix D.1 for results using usage-specific axes alone.

**Analysis** As shown in Table 1, integrating usage-specific sub-axes with the main sentiment axis (Main+Sub) yields an average gain of +4.60% over the main-axis baseline across eight datasets. The effect, however, varies substantially across datasets and models. **At the dataset level**, the largest gains appear on subjective and stylistically diverse corpora such as Confession, Cringe, and OkCupid, where sentiment is often mediated by genre, discourse tone, or contextual cues; by contrast, SST5, with its short and explicit sentiment expressions, leaves little room for usage factors to contribute, so their signal is relatively weak and may even lead to minor performance fluctuations for LLaMA (-2.0%). **At the model level**, Gemma shows the largest benefit (+6.5% on average) due to its weaker main axis, whereas LLaMA and Mistral, with stronger main axes, gain smaller but steady improvements. Importantly, usage-specific sub-axes also tend to act as a leveling mechanism: across datasets such as SST5, Twitter, Dialogue, Animals, Conf, and OkC, models with different baseline strengths converge to a much narrower accuracy range once sub-axes are incorporated. For instance, on Confession, the three models differ by 7% in the baseline setting, but converge to 74.7–77.7% after adding usage-specific sub-axes.

Table 2 shows that the optimal Main+Sub combinations vary across datasets and models. Context is consistently helpful for long reviews (IMDb), topic is favored for short posts such as Twitter, and genre plays a key role in dialogues and user-generated content (Conf, Cringe, OkCupid). Animals benefit from combining genre and topic. Across models, LLaMA often relies on multi-factor combinations, Gemma balances between single- and multi-axis choices, and Mistral shows a strong tendency to incorporate topic, including the only four-axis combination observed on Conf. These patterns highlight that usage-aware axes not only improve accuracy but also reveal interpretable variation across models and datasets in how sentiment is encoded.

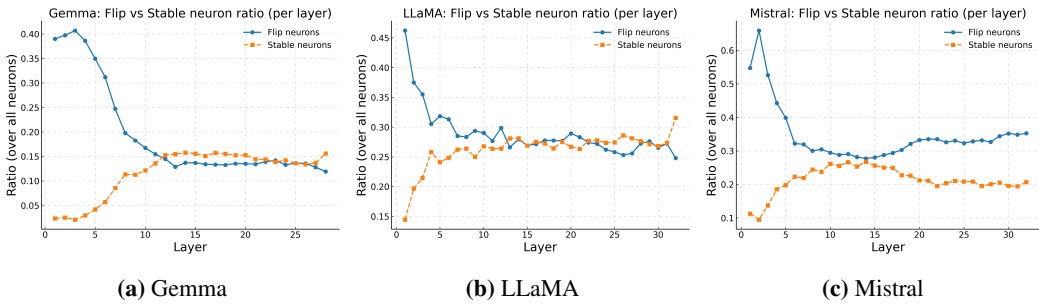

**(a)** Gemma       **(b)** LLaMA       **(c)** Mistral

**Figure 3:** Per-layer ratios of flip and stable neurons over all neurons for three models. Each curve shows the fraction of neurons of the given type at each layer.

## 4.2 NEURON-LEVEL PROBING

To understand why usage-specific sub-axes improve performance, we conduct a neuron-level probe (Section 3) that measures how individual neurons vary in sentiment polarity across usages.

**Training Probes** Building on the heterogeneity observed in Figure 2, we distinguish two neuron subsets: stable neurons, whose polarity remains consistent across usages, and flipped neurons, whose polarity changes across usages. To assess their roles in sentiment encoding, we design two complementary probes. (i) Masking analysis: we selectively mask stable or flipped neurons along the main sentiment axis to test their contribution to existing axes. (ii) Retraining analysis: we retrain sentiment axes using only stable or flipped subsets to examine how each group independently encodes sentiment and how they interact when combined. Together, these analyses show that sentiment representation is distributed across both types: stable neurons form an invariant backbone, while flipped neurons act as usage-sensitive modulators.

**Layer-wise Distribution** Figure 3 shows the proportion of flip and stable neurons across layers for the three LLMs. A consistent trend emerges: flip neurons dominate in the shallow layers but gradually give way to stable neurons as depth increases. This pattern likely arises because shallow layers encode lexical and surface-level cues, such as tone markers or topic-specific words, that are strongly modulated by usage, whereas deeper layers abstract away from these idiosyncratic signals and consolidate more usage-invariant sentiment features. While this general pattern holds across models, they differ in emphasis: Gemma exhibits an earlier peak in flip activity, Mistral sustains a relatively high flip ratio across layers, and LLaMA displays a smoother transition toward stability.

**Table 3:** Accuracy (%) when the main axis is masked: performance of Main, Flip, Stable, and Flip+Stable subsets across models and datasets.

| Dataset | Gemma | | | | LLaMA | | | | Mistral | | | |
|---|---|---|---|---|---|---|---|---|---|---|---|---|
| | Main | Flip | Stable | Flip+Stable | Main | Flip | Stable | Flip+Stable | Main | Flip | Stable | Flip+Stable |
| IMDb | 78.38 | 64.18 | 78.56 | **83.70** | 78.15 | 58.79 | 84.06 | **85.43** | 74.22 | 61.38 | 80.30 | **83.26** |
| SST5 | 79.58 | 59.37 | 79.00 | **83.06** | 83.98 | 59.58 | 83.67 | **84.23** | 77.02 | 59.08 | 75.71 | **84.26** |
| Twitter | 83.06 | 67.34 | 86.25 | **88.81** | 87.95 | 68.27 | 90.66 | **91.24** | 88.66 | 69.47 | 88.40 | **89.98** |
| Dialogue | 79.11 | 78.26 | 87.79 | **91.47** | 87.40 | 78.42 | 89.02 | **91.78** | 86.79 | 78.80 | 86.25 | **89.55** |
| Animals | 74.86 | 76.00 | 80.57 | **87.43** | 82.86 | 76.57 | 83.43 | **85.71** | 81.14 | 79.43 | 81.14 | **85.14** |
| Conf | 65.88 | 65.88 | 73.53 | **80.59** | 71.76 | 67.65 | 75.29 | **78.82** | 72.94 | 64.12 | 75.29 | **78.24** |
| Cri | 61.70 | 62.23 | 69.15 | **75.00** | 62.23 | 62.23 | 69.68 | **75.53** | 68.09 | 61.70 | 71.28 | **75.53** |
| OkC | 63.52 | 69.18 | 72.96 | **79.25** | 72.33 | 68.55 | 76.73 | **79.87** | 69.81 | 66.67 | 71.70 | **80.50** |

**Analysis** Tables 3 and 4 provide complementary views on the roles of stable and flipped neurons. Importantly, the flipped directions reported here are not raw signals, but have been adjusted with usage-aware signs ($\pm 1/0$) based on usage-specific statistics (positive–negative activation differences). This adjustment prevents polarity reversals across usages from canceling out, aligning flipped neurons into consistent conditional features. In the masked setting (Table 3), stable neurons generally outperform the full main axis, indicating that they provide a cleaner and more robust backbone by filtering out misaligned signals. Flip-only results, though usage-sensitive, remain weaker: flipped neurons provide conditional cues that are effective within specific usages but lack robustness when aggregated across diverse test domains. When combined with stable neurons, however,

flipped neurons contribute complementary benefits: Flip+Stable achieves the best results on several datasets. This pattern shows that stable neurons capture invariant sentiment signals, while usage-aware flipped neurons enrich sentiment boundaries by encoding usage-dependent variation.

In the retrained setting (Table 4), flipped neurons achieve stronger performance within their own subspace, narrowing the gap with stable neurons. This shows that flipped neurons also encode sentiment-discriminative signals, but only when usage variation is explicitly modeled. Overall, sentiment in LLMs appears to emerge from the interplay between stable neurons, which provide robustness, and flipped neurons, which adapt to usage differences. For completeness, we additionally evaluate all sixteen possible usage combinations; detailed results are reported in Appendix D.2.

**Table 4:** Accuracy (%) when retraining axes within each neuron subset: performance of Main, Flip, Stable, and Flip+Stable across models and datasets.

| Dataset | Gemma | | | | LLaMA | | | | Mistral | | | |
|---|---|---|---|---|---|---|---|---|---|---|---|---|
| | Main | Flip | Stable | Flip+Stable | Main | Flip | Stable | Flip+Stable | Main | Flip | Stable | Flip+Stable |
| IMDb | 78.38 | 76.96 | 78.01 | **83.72** | 78.15 | 81.68 | 78.04 | **82.26** | 74.22 | 75.35 | 78.14 | **83.54** |
| SST5 | 79.58 | 76.31 | 78.68 | **80.83** | **83.98** | 76.92 | 79.42 | 82.41 | 77.02 | 74.85 | 77.73 | **82.59** |
| Twitter | 83.06 | 80.67 | 83.76 | **84.72** | 87.95 | 75.69 | 89.68 | **90.58** | 88.66 | 76.26 | 88.97 | **89.82** |
| Dialogue | 79.11 | 83.72 | 83.56 | **89.25** | 87.40 | 81.26 | 90.17 | **90.94** | 86.79 | 77.88 | 88.56 | **90.63** |
| Animals | 74.86 | 78.29 | 78.86 | **85.71** | 82.86 | 79.43 | 85.14 | **85.14** | 81.14 | 76.00 | 81.71 | **83.43** |
| Conf | 65.88 | 71.18 | 68.82 | **75.29** | 71.76 | 71.76 | 71.76 | **78.24** | 72.94 | 68.24 | 76.47 | **78.82** |
| Cri | 61.70 | 69.68 | 68.63 | **70.74** | 62.23 | 65.43 | 68.62 | **74.47** | 68.09 | 68.09 | 70.21 | **75.00** |
| OkC | 63.52 | 69.18 | 67.92 | **74.84** | 72.33 | 69.81 | 74.21 | **80.50** | 69.81 | 69.81 | 70.44 | **76.10** |

## 5 SENTIMENT STEERING WITH LEARNED DIRECTIONS

To investigate whether learned sentiment directions can be used for controllable generation, we conduct intervention experiments using LLaMA as the generation model, applying the inference-time activation modification approach described in Section 2. Since sentiment in natural language is rarely expressed in isolation—a review, a dialogue, or a news report may convey the same polarity through different stylistic conventions—we test whether generation can be steered along usage-conditioned directions (tone, genre, context, topic) beyond simple polarity flips.

**Usage-axis Steering** We test whether usage-conditioned sentiment directions enable controllable generation. Interventions are applied at layer 14 along four usage axes (tone, genre, context, topic), with the main axis disabled ($\alpha = 0$, $\beta \in \pm 30\sigma$). For each usage we compare two variants: raw, the learned direction, and ortho, obtained after removing its projection onto the main sentiment axis. As shown in Table 5, raw directions remain closely aligned with sentiment polarity and primarily flip when $\beta$ changes sign, whereas orthogonalized directions no longer affect polarity but instead modulate framing and style. Tone and topic exhibit clear shifts (e.g., from frustration to engaged discussion, or from conflict to monotony), while genre and context yield subtler effects, shifting emphasis from outcome evaluations toward process- and atmosphere-related descriptions. For completeness, Appendix D.3 reports results when steering only with the main axis, which reliably controls sentiment polarity. This contrast highlights that while the main axis captures polarity, usage-conditioned axes further regulate stylistic variation in sentiment expression.

**Quantitative Effects of Usage** We steer hidden states at layer 14 along four usage directions (tone, genre, context, topic) using 50 neutral prompts (Appendix F). The intervention is defined as $x'_{\ell,h} = x_{\ell,h} + \alpha \, \mathrm{RMS}(h) \, \hat{w}_{\mathrm{main}} + \beta \, \mathrm{RMS}(h) \, \hat{w}_{\mathrm{usage}}$, tested under two conditions: usage-only ($\alpha = 0$, $\beta \in \{-30, -15, 0, 15, 30\}$) and main+usage ($\alpha = 10$, $\beta \in \{-40, -30, -15, 0, 15, 30, 40\}$). We compare raw usage directions with their orthogonalized versions, where overlap with the main sentiment axis is removed. Figure 4a shows a clear dose–response: increasing $\beta$ strengthens polarity shifts, though the magnitude differs by usage. Figure 4b quantifies this with average slopes of $\beta \to \Delta s$. Orthogonalization attenuates all directions, but genre declines most sharply, reflecting its strong alignment with the main sentiment axis (cosine 0.77 vs. 0.56–0.65 for the others). These findings demonstrate that sentiment in LLMs is not encoded along a single universal axis. Instead, sentiment is usage-aware: each usage direction exhibits a measurable cosine similarity with the main axis, yet they are not redundant, as orthogonalization leaves nontrivial residuals. Moreover,

| Usage | $\beta = -30\sigma$ | | $\beta = +30\sigma$ | |
|---|---|---|---|---|
| | **Raw** | **Ortho** | **Raw** | **Ortho** |
| **Tone** | *no resolution; frustrated and exhausted; no progress* | *intense discussion; plan found; answer emerged* | *productive meeting; energy and positivity; gratitude* | *heated yet focused; challenges managed* |
| **Genre** | *long and tedious; heavy boredom; glacial pace* | *marathon session; tense room; fatigue* | *brainstormed ideas; energized and motivated* | *various topics; breaks; productive after all* |
| **Context** | *waste of time; no agreement; unproductive* | *no outcome; frustration and tension* | *productive and engaging; inspired; progress made* | *laughter and ideas; wonderful time* |
| **Topic** | *repetition; no clear resolution; tempers fray* | *hard to stay focused; monotone; slow clock* | *excited to discuss; successful outcome; celebration* | *engaged and enthusiastic; inspiring session* |

**Table 5:** Qualitative outputs from usage-only steering at layer 14 ($\alpha = 0$, $\beta = \pm30\sigma$). Raw denotes the learned usage direction, which primarily flips polarity when $\beta$ changes sign, while Ortho denotes the same direction after removing its projection onto the main sentiment axis, mainly shifting framing/style. Prompt: "The meeting lasted for two hours. Write about this situation." Full results are in Appendix E.

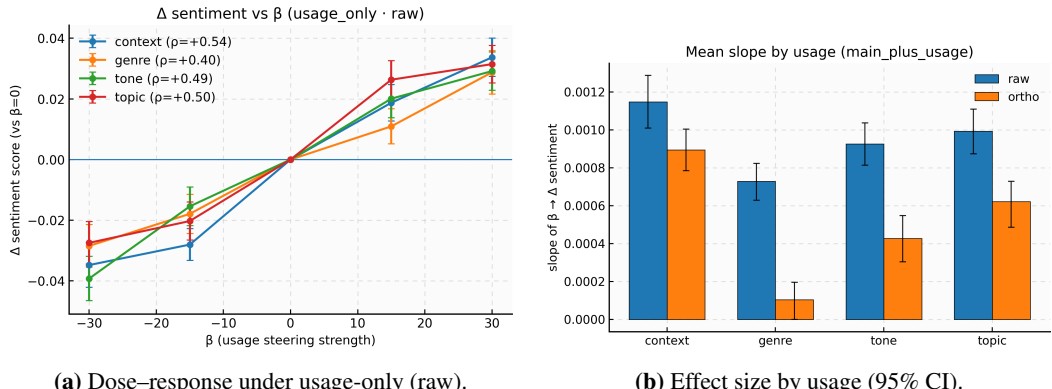

**(a)** Dose–response under usage-only (raw).      **(b)** Effect size by usage (95% CI).

**Figure 4:** Quantifying steering by usage. (a) Baseline-corrected sentiment $\Delta s$ versus usage strength $\beta$ (mean $\pm$ SEM) under usage-only (raw). Legend in the plot reports Spearman $\rho$. (b) Mean slope of $\beta \to \Delta s$ across prompts under main+usage steering; orthogonalization (orange) reduces magnitude, especially for genre.

combining main with usage directions yields higher classification accuracy than the main axis alone, confirming that usage factors provide complementary contributions to sentiment representation.

At layer 14 of LLaMA, we asked how much of the pooled main sentiment axis can be explained by usage factors (genre, tone, context, topic). To test this, we projected the main axis into the four-dimensional usage span and measured how much variance was captured. We found that about 86% of the main axis lies within the usage subspace, indicating that most polarity signal is explained by usage-specific factors.

**How Much of the Main Axis Lies in Usage?** The small residual (14%) instead captures lexical and narrative modulations beyond polarity, suggesting that the main axis is not a pure sentiment direction but largely shaped by usage-conditioned variation.

We next steer along the residual main⊥usage direction. In Figure 5, once the usage-aligned component is removed, the scale's sign no longer maps polarity in a consistent way: in the illustrated example, $-\alpha$ produces a more positive description than $+\alpha$.

This sign instability suggests that the polarity signal of the main axis is primarily carried by its usage-aligned component, while the residual instead governs stylistic variations in a non-monotonic way.

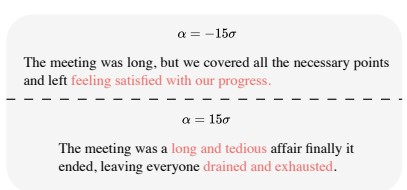

**Figure 5:** Sign instability after removing usage. On main ⊥ usage, flipping the scale sign no longer deterministically maps to polarity, but instead interacts with context in a non-monotonic way.

## 6    EXTENDING USAGE-AWARE SENTIMENT AXES TO SPEECH

We further examine whether our findings extend to speech. Using a synthetic audio dataset generated with the CosyVoice TTS system (Du et al., 2024) and annotated with usage dimensions (prosody, context, topic, genre), we train main and usage-specific sentiment axes following the same probing procedure as for text. We evaluate the learned axes on the IEMOCAP dataset (Busso et al., 2008) (Session 5 test split) under a binary sentiment classification setup. The main axis achieves 76.12% accuracy, while combining it with usage-specific axes further improves performance to 78.69% (best layer 3). This +2.57 pp gain is statistically reliable: bootstrap resampling gives a 95% confidence interval of [0.8, 4.3], and McNemar's test indicates significance (p = 0.0038, b = 74, c = 42). The fact that the best result occurs in an early layer is consistent with speech emotion being strongly tied to low-level prosodic cues. These results confirm that sentiment in speech is not monolithic but intertwined with usage factors. Full details and extended results are provided in Appendix G.

## 7    RELATED WORK

Probing has emerged as a central methodology for analyzing what kinds of information are encoded within neural network representations. Early work introduced linear probes as lightweight classifiers for diagnosing how information is distributed across layers of a network (Alain & Bengio, 2016). This basic approach was soon extended beyond generic representations to capture linguistic structures such as syntax and morphology, offering a way to test whether networks internalize hierarchical linguistic patterns (Hewitt & Manning, 2019; Belinkov, 2022). These efforts led to what is now known as the linear representation hypothesis, which posits that high-level semantic and conceptual attributes correspond to approximately linear directions in the latent space of large models (Park et al., 2024; Jiang et al., 2024; Mikolov et al., 2013). Building on this foundation, researchers have explored an increasingly broad set of conceptual attributes in large language models (LLMs). Probing directions have been identified for domains as diverse as political stance (Kim et al., 2025), refusal behaviors (Arditi et al., 2024), perceptual properties such as color (Patel & Pavlick, 2022), spatiotemporal reasoning (Gurnee & Tegmark, 2024), persona construction (Chen et al., 2025), detoxification and harmful content mitigation (Turner et al., 2023), and appropriateness or politeness in dialogue (Von Rütte et al., 2024). Sentiment has likewise been studied under this framework (Tigges et al., 2024). However, a recurring challenge is that probe-derived directions for sentiment often vary substantially across datasets, raising concerns that the discovered axes may reflect dataset-specific artifacts rather than robust, generalizable properties of language models. To address such instability, distributional approaches have been proposed. For instance, Gaussian subspace methods (Zhao et al., 2025) represent conceptual attributes not with a single axis but with a set of axes capturing a distribution of latent directions. While these methods can mitigate sensitivity to dataset variation, they introduce a significant drawback: interpretability suffers because the learned representations are no longer aligned with simple, human-understandable directions. In contrast to these approaches, our work contributes a probing methodology that employs a small set of linguistically informed, usage-based axes. By grounding probing directions in how sentiment is actually expressed in language, we improve interpretability while still maintaining robustness to dataset variation. This design choice extends prior work by striking a balance between the clarity of linear probing and the stability of distributional approaches, ultimately providing a framework better suited for analyzing sentiment in LLMs.

## 8    CONCLUSIONS

In this work, we introduced a usage-aware sentiment representation framework that accounts for sentiment variability by explicitly modeling linguistic factors such as tone, topic, context, and genre. Unlike prior distributional approaches, our method leverages a small, interpretable set of usage-based directions. At the axis level, we construct sentiment directions from both pooled and usage-specific data, while at the neuron level we distinguish usage-invariant from usage-sensitive neurons, offering a finer-grained view of how sentiment is encoded. Through extensive experiments, we show that this framework yields more reliable sentiment directions, improves interpretability, and strengthens downstream applications, supporting cross-domain sentiment classification and enabling controllable generation via sentiment steering. Finally, preliminary results with audio LLMs suggest that the framework generalizes beyond text, pointing toward broader cross-modal applicability.

## ETHICS STATEMENT

This work studies usage-awaresentiment representations in large language models. All datasets used are either synthetically generated or publicly available benchmarks, and thus do not involve private or personally identifiable information. While usage-aware sentiment modeling can improve reliability and interpretability, sentiment steering also raises potential risks of misuse, such as manipulating emotions or amplifying bias. Our methods are intended purely for research purposes and should not be applied directly in sensitive domains such as clinical mental health support. We believe our study contributes to responsible interpretability research by clarifying how usage factors shape sentiment in LLMs, while highlighting the need for caution in downstream applications.

## REPRODUCIBILITY STATEMENT

We have made substantial efforts to ensure the reproducibility of our results. All code for data preprocessing, model training, and evaluation is provided in the supplementary materials.

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

## A USAGE-ANNOTATED DATA

We generate a synthetic usage-annotated dataset covering four categories—tone, genre, context, and topic—each with multiple sub-types (Table 6). For every usage type, we construct both positive and negative sentiment prompts, yielding 2,000 samples per category (1,000 positive and 1,000 negative). Table 7 summarizes dataset statistics, showing that all four categories are sentiment-balanced but differ in average prompt length: tone prompts are shortest, whereas genre prompts are the longest. As illustrative examples, we provide the full set of prompt templates for each category: Table 8 (tone), Table 9 (genre), Table 10 (context), and Table 11 (topic). All four categories follow the same design pattern, ensuring consistency across usage factors.

| Usage Category | Types |
|---|---|
| Tone | Sincere, Sarcastic, Excited, Formal, Casual |
| Genre | Romantic Fiction, Horror Narrative, Satirical Comedy, Motivational Speech, Melancholic Poetry |
| Context | Product Review, Movie Review, Political Opinion, Travel Experience, Tech News |
| Topic | Politics, Finance, Health, Relationships, Sports |

**Table 6:** Overview of usage categories and their corresponding types.

| Usage Dimension | Samples | Avg. Length (tokens) | Pos:Neg | Ratio |
|---|---|---|---|---|
| Tone | 2000 | 63.34 | 1000:1000 | 1.0 |
| Context | 2000 | 131.67 | 1000:1000 | 1.0 |
| Genre | 2000 | 182.67 | 1000:1000 | 1.0 |
| Topic | 2000 | 156.95 | 1000:1000 | 1.0 |

**Table 7:** Statistics of generated usage-based sentiment datasets. Each dimension contains 2000 samples (balanced with 1000 positive and 1000 negative).

| Usage Type | Positive Prompts | Negative Prompts |
|---|---|---|
| Sincere | - Write a sincere sentence expressing gratitude or joy.
- Say something heartfelt to a friend who helped you. | - Write a sincere sentence expressing regret or apology.
- Say something honest and remorseful after a failure. |
| Sarcastic | - Write a sarcastic sentence that ironically praises something.
- Say something positive in a clearly sarcastic tone. | - Write a sarcastic remark criticizing someone or something.
- Make a bitter joke about a frustrating experience. |
| Excited | - Write an excited sentence about achieving something great.
- Show your enthusiasm after a big success. | - Write an excited-sounding complaint about something terrible.
- Express extreme frustration in an over-the-top tone. |
| Formal | - Write a formal sentence commending someone's professional performance.
- Express official recognition of a project's success. | - Write a formal sentence delivering disappointing news.
- Politely inform someone of a failed application. |
| Casual | - Write a casual, happy sentence about a great day.
- Say something fun and laid-back after something good happened. | - Write a casual sentence complaining about something annoying.
- Say something informal about a disappointing situation. |

**Table 8:** Prompt templates for tone-based usage data generation. Each usage type has both positive and negative sentiment prompts.

| Usage Type | Positive Prompts | Negative Prompts |
|---|---|---|
| Romantic Fiction | - Write a short love letter expressing happiness and affection.
- Describe a romantic moment that made you feel joyful. | - Write a short love letter expressing heartbreak and sadness.
- Describe a painful romantic experience that left you feeling down. |
| Horror Narrative | - Describe the moment a horror story character finally escaped to safety.
- Write a scene where survivors realize the danger has passed and feel relieved. | - Write a chilling scene filled with fear and suspense.
- Describe a horror moment where the protagonist feels hopeless. |
| Satirical Comedy | - Write a funny sarcastic sentence that ends up praising something.
- Create a light-hearted satirical compliment about a tech product. | - Write a sarcastic tweet criticizing airline service.
- Mock a politician using humor and irony to express frustration. |
| Motivational Speech | - Write a motivational sentence that inspires confidence and pride.
- Encourage someone who feels uncertain with a positive message. | - Describe a moment of self-doubt or fear in a motivational speech.
- Write a sentence about the pain of failure before recovery. |
| Melancholic Poetry | - Compose a four-line poem expressing peace after letting go.
- Write a short poem about gratitude after a loss. | - Compose a melancholic poem about deep sorrow and loneliness.
- Write a short elegy mourning a lost friend. |

**Table 9:** Prompt templates for **genre**-based usage data generation.

| Usage Type | Positive Prompts | Negative Prompts |
|---|---|---|
| Product Review | - Write a short positive review of a consumer product.
- Describe what you loved about a product you recently bought. | - Write a short negative review of a disappointing product.
- Describe why a product you bought recently let you down. |
| Movie Review | - Write a short, enthusiastic review of a great movie you've seen.
- Describe what impressed you most in a recent film. | - Write a short negative reaction to a bad movie.
- Describe why a film you watched was disappointing. |
| Political Opinion | - Share a positive opinion about a recent political action or policy.
- Write a short message supporting a political leader or idea. | - Write a short critical comment about a political situation.
- Express your frustration with a recent government decision. |
| Travel Experience | - Share a joyful moment from your recent travel.
- Describe a highlight from a great trip. | - Describe a disappointing travel experience in one sentence.
- Write a short complaint about a travel destination. |
| Tech News | - Write a short excited reaction to a new tech product.
- Describe something great about a recent tech launch. | - Write a negative opinion about a new tech product.
- Share a complaint about a feature in a new device. |

**Table 10:** Prompt templates for **context**-based usage data generation.

| Usage Type | Positive Prompts | Negative Prompts |
|---|---|---|
| Politics | - Write a short statement praising a new progressive policy.
- Describe a positive social change driven by recent legislation.
- Write a hopeful line about a promising political movement. | - Write a short remark criticizing a recent government decision.
- Describe frustration about political inaction on important issues.
- Write a line expressing disappointment in leadership. |
| Finance | - Describe the joy of achieving a financial goal.
- Write a short message about a successful investment outcome.
- Share excitement about receiving a salary raise or bonus. | - Write a sentence about struggling with debt or bills.
- Describe the stress of losing money on an investment.
- Write a short complaint about rising living costs. |
| Health | - Share a joyful moment after getting a clean health report.
- Describe the feeling of completing a fitness goal.
- Write a line about emotional recovery and renewed energy. | - Write about the pain of living with chronic illness.
- Describe the anxiety of waiting for medical test results.
- Write a sentence about losing motivation due to poor health. |
| Relationships | - Write a sentence about reuniting with someone you love.
- Describe a moment of closeness in a healthy relationship.
- Share joy from spending time with family or friends. | - Write a line about feeling lonely in a relationship.
- Describe a recent argument that left emotional pain.
- Write a short reflection on falling out with a friend. |
| Sports | - Describe the thrill of winning a big game.
- Write a celebratory sentence after setting a personal record.
- Share excitement after a team's comeback victory. | - Write about the frustration of losing in the final round.
- Describe disappointment after a key player got injured.
- Write a short line about failing to qualify in a tournament. |

**Table 11:** Prompt templates for **topic**-based usage data generation.

## B  TEST DATA

As shown in Table 12, we use SST5 in its binary form, where neutral instances are removed and the remaining labels are collapsed into positive and negative categories. Our usage-aware sentiment representations achieve consistently strong performance across all evaluation datasets, demonstrating robustness in diverse out-of-domain settings.

| Dataset | Total | Positive | Negative |
|---|---|---|---|
| AnimalsBeingBros (Reddit) | 175 | 128 | 47 |
| Confession (Reddit) | 170 | 102 | 68 |
| Cringe (Reddit) | 188 | 104 | 84 |
| OkCupid (Reddit) | 159 | 102 | 57 |
| DailyDialog | 1302 | 1019 | 283 |
| SST5 (binary phrases) | 26052 | 14789 | 11263 |
| IMDb (test) | 25000 | 12500 | 12500 |
| Twitter (TweetEval, binary) | 5151 | 3693 | 1458 |

**Table 12:** Statistics of evaluation datasets. For Twitter (TweetEval), we report the binary subset after removing neutral instances (label=1).

## C  MODELS

We use instruction-tuned versions of three popular decoder-only LLMs. We use instruction-tuned versions of three popular decoder-only LLMs (Gemma-7B-IT, LLaMA-3-8B-Instruct, Mistral-7B-Instruct). We choose instruction-tuned models because they represent the most widely used variants in real-world applications, where robustness to sentiment and controllability of outputs are especially relevant. Using IT versions also ensures comparability across models, since their pretraining and fine-tuning objectives are aligned toward instruction following.

- **LLaMA-3-8B-Instruct**: Meta's instruction-tuned model with 8.0B parameters, 32 layers, hidden size 4096, 32 attention heads, and 8k context window.
- **Gemma-7B-Instruct**: Google's 7B parameter instruction-tuned model with 28 layers, hidden size 3072, 16 attention heads, and 8k context window, released under the Gemini program.
- **Mistral-7B-Instruct**: A 7.3B parameter dense transformer with 32 layers, hidden size 4096, 32 attention heads, and 8k context window, featuring grouped-query and sliding-window attention.

All models are accessed through HuggingFace Transformers. For probing, we freeze the backbone weights and train only logistic regression probes.

# D ADDITIONAL RESULTS

## D.1 USAGE-SPECIFIC RESULTS

Figure 6 shows the complete per-dataset results considering only usage-specific axes, illustrating how the effectiveness of different usage factors varies across datasets. The plots make explicit which usage dimensions (e.g., topic, genre, context, tone) provide the strongest complement to the main sentiment axis in each setting.

We observe that the best-performing single usage axis often aligns with the optimal Main+Sub combination. For example, on SST5 with Mistral-7B-Instruction, the topic axis achieves the highest standalone performance, and the Main+Topic combination also yields the best overall accuracy. Similar patterns occur across several datasets, indicating that the most informative usage dimension typically dominates the combined representation. Overall, incorporating usage axes improves performance, though the size of the gain depends on both the dataset and the model.

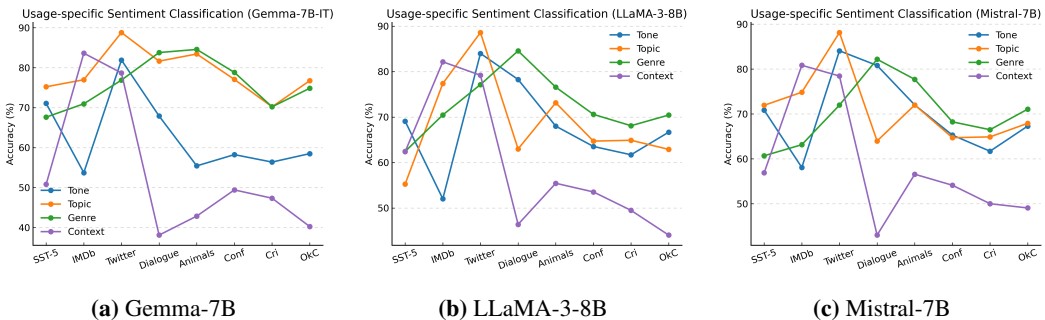

(a) Gemma-7B  (b) LLaMA-3-8B  (c) Mistral-7B

**Figure 6:** Usage-specific sentiment classification results across three LLMs. Each subfigure shows accuracy across eight cross domain datasets.

## D.2 NEURON RESULTS ACROSS USAGE COMBINATIONS

Figures 7–12 summarize the accuracy of sentiment classification under different neuron-level usage combinations. Figures 7–9 correspond to the *masking design*, where axes are restricted to stable or flipped neurons and their combinations. Figures 10–12 correspond to the *retrained setting*, where models are fine-tuned within the flip+stable subspace. In both settings, results are reported across datasets and usage combinations, with labels showing the best-performing layer and accuracy. The x-axis letters denote usage types: **g** = genre, **t** = topic, **T** = tone, **c** = context, and ∅ = no usage factor. The best-performing combination for each dataset is highlighted in gold.

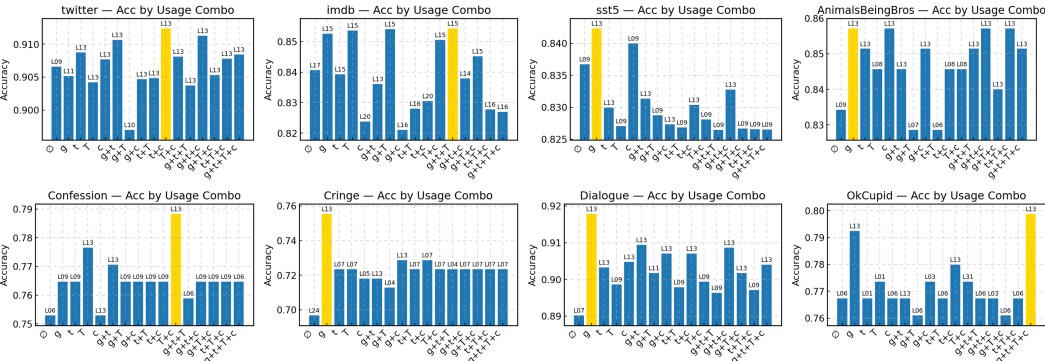

**Figure 7:** Sentiment classification accuracy for **LLaMA** under the *masking design*. Each bar denotes a usage-specific axis, with labels showing the best-performing layer and accuracy. The best usage combination for each dataset is highlighted in gold.

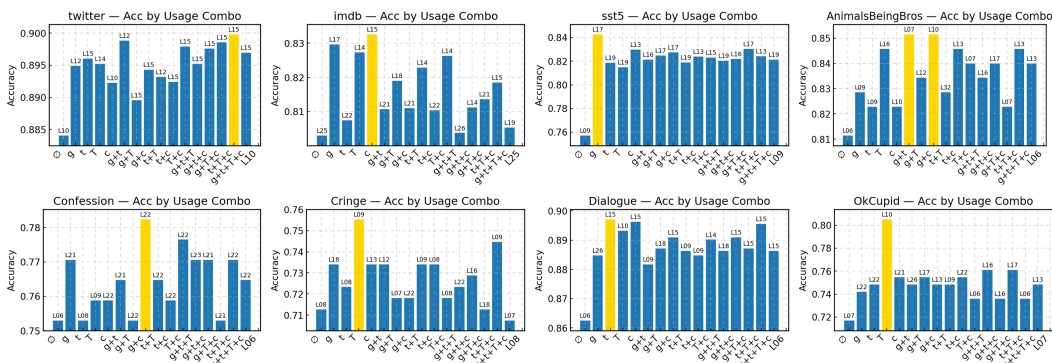

**Figure 8:** Sentiment classification accuracy for **Mistral** under the *masking design*. Each bar denotes a usage-specific axis, with labels showing the best-performing layer and accuracy. The best usage combination for each dataset is highlighted in gold.

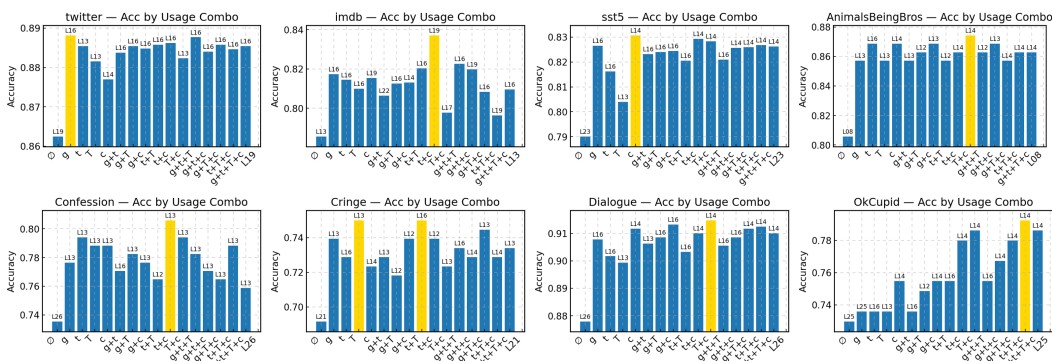

**Figure 9:** Sentiment classification accuracy for **Gemma** under the *masking design*. Each bar denotes a usage-specific axis, with labels showing the best-performing layer and accuracy. The best usage combination for each dataset is highlighted in gold.

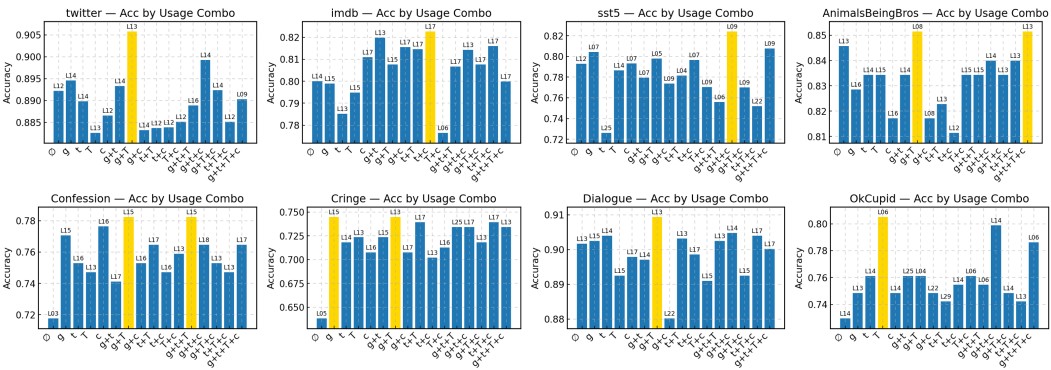

**Figure 10:** Sentiment classification accuracy for **LLaMA** in the *retrained flip+stable* setting (Table 4). Each bar corresponds to a dataset, with labels showing the best-performing layer. The best accuracy for each dataset is highlighted in gold.

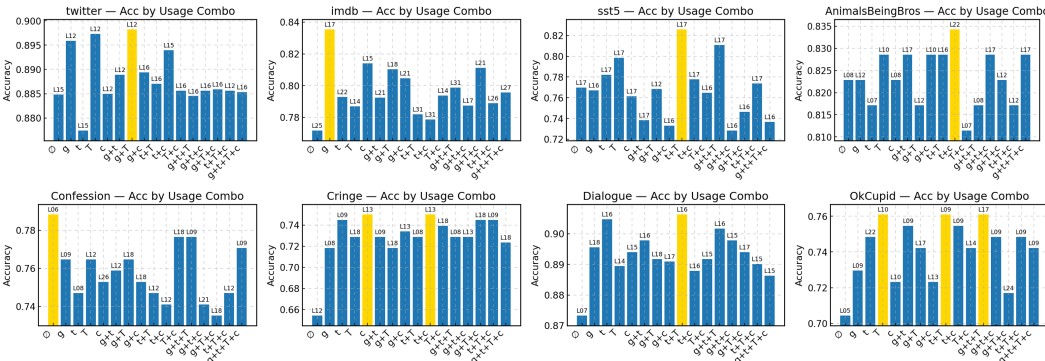

**Figure 11:** Sentiment classification accuracy for **Mistral** in the *retrained flip+stable* setting (Table 4). Each bar corresponds to a dataset, with labels showing the best-performing layer. The best accuracy for each dataset is highlighted in gold.

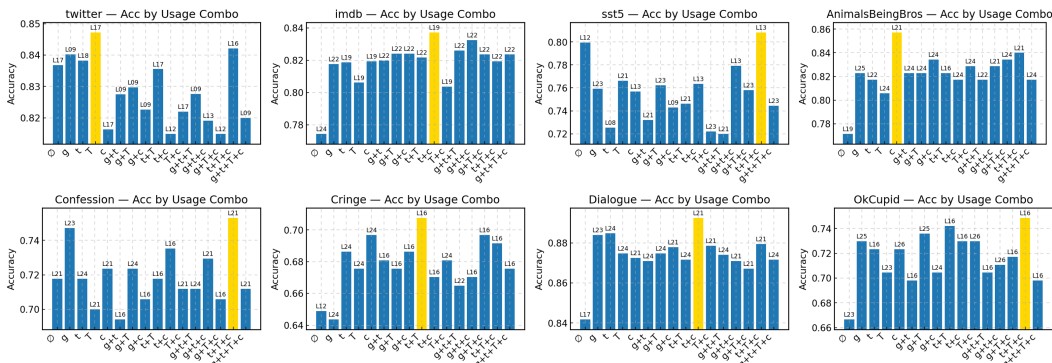

**Figure 12:** Sentiment classification accuracy for **Gemma** in the *retrained flip+stable* setting (Table 4). Each bar corresponds to a dataset, with labels showing the best-performing layer. The best accuracy for each dataset is highlighted in gold.

### D.3 Additional Steering Examples

Figure 13 and Table 13 together illustrate controllable generation along the main sentiment axis at layer 14. The prompt is fixed as:

> *"A person waits in a hospital corridor for test results. Write ∼60 words in third person, present tense."*

While extreme scalings ($\alpha = \pm 30\sigma$) produce clearly anxious vs. hopeful language, the intermediate sweep ($\alpha = -15\sigma, 0, +15\sigma$) reveals an interesting asymmetry: the zero point does not correspond to neutral text, but instead retains anxious phrasing. This suggests that the model's baseline representation of the prompt already encodes a bias toward negative affect in this scenario, rather than starting from a truly neutral state.

| $\alpha$ | Example Generation |
|---|---|
| $-15\sigma$ | "The doctor is expected to arrive soon with the news. As she sits in the hospital corridor, her eyes fixed on the clock, Sarah's anxiety grows with each passing minute. She has been waiting for what feels like an eternity for the doctor to deliver the test results, her mind racing with worst-case scenarios. The minutes tick by slowly as she tries to distract herself from the uncertainty." |
| $0\sigma$ | "The doctor is expected to arrive soon with the news. As she sits in the hospital corridor, her eyes fixed on the clock, Sarah's anxiety grows. She has been waiting for what feels like an eternity for the doctor to deliver the test results that will determine her future. The minutes tick by slowly as she tries to calm her racing thoughts and prepare herself for whatever news may come." |
| $+15\sigma$ | "The patient's anxiety grows as the minutes tick by. They fidget with their hands, glancing at the clock every few seconds. Their mind races with worst-case scenarios, but they try to focus on the positive outcomes. As the wait continues, their breathing slows and their body relaxes, preparing them for whatever news is to come." |

**Table 13:** Generations under intermediate $\alpha$ strengths along the main sentiment axis at layer 14. Negative scaling induces more anxious phrasing, while positive scaling yields calmer and more hopeful language.

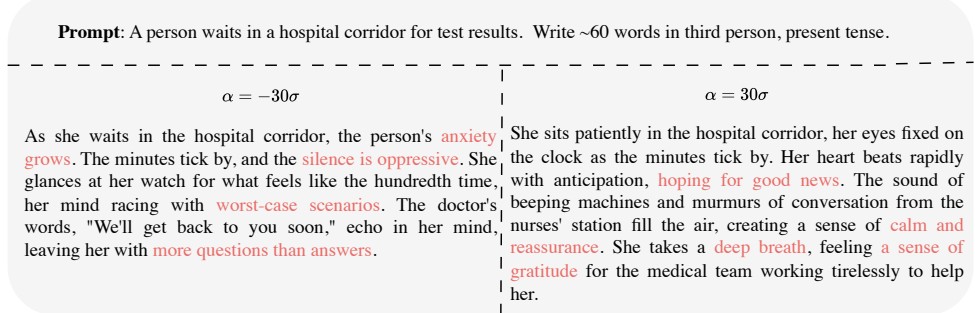

**Figure 13:** Controllable generation by steering the main sentiment axis at layer 14: negative scaling ($\alpha = -30\sigma$) yields anxious text, while positive scaling ($\alpha = +30\sigma$) produces hopeful language.

# E  FULL RESULTS OF USAGE-ONLY STEERING AT LAYER 14

This appendix presents complete qualitative results of usage-only steering ($\alpha = 0, \beta = \pm 30\sigma$) at layer 14. Each table shows Raw (usage axis only) vs. Ortho (orthogonalized).

TONE

| $\beta$ | Raw | Ortho |
|---|---|---|
| $+30\sigma$ | The meeting was productive and filled with energy and positivity. Everyone felt grateful for the collaboration. | The meeting was long and tiring, with heated discussions, yet the team managed to stay focused and motivated. |
| $-30\sigma$ | A planned short meeting dragged on for hours, leaving everyone frustrated and exhausted, with no resolution. | The discussion was intense but eventually led to a workable plan and an answer. |

GENRE

| $\beta$ | Raw | Ortho |
|---|---|---|
| $+30\sigma$ | The meeting was productive; ideas were brainstormed and participants left energized and motivated. | The meeting covered various topics with breaks, but was ultimately productive after all. |
| $-30\sigma$ | The meeting was long and tedious, dragging at a glacial pace with heavy boredom. | The session felt like a marathon, with a tense room and fatigue dominating. |

CONTEXT

| $\beta$ | Raw | Ortho |
|---|---|---|
| $+30\sigma$ | The meeting was productive and engaging, leaving participants inspired and motivated. | The atmosphere was filled with laughter, ideas, and inspiration; a wonderful time was had. |
| $-30\sigma$ | The meeting felt like a waste of time; no agreement was reached and it remained unproductive. | Despite efforts, the outcome was frustration and tension, with no meaningful progress. |

TOPIC

| $\beta$ | Raw | Ortho |
|---|---|---|
| $+30\sigma$ | The team eagerly discussed their project, celebrated success, and felt inspired by the outcome. | The agenda covered strategies and reports; participants were engaged and enthusiastic throughout. |
| $-30\sigma$ | The meeting was repetitive, with no clear resolution, and tempers began to fray. | Endless discussions dragged on monotonously, with no resolution in sight. |

## F  STEER PROMPTS

**Table 14:** Steering Prompts (each used to generate ∼60 words in third person, present tense)

| # | Prompt |
|---|---|
| 1 | The meeting lasts for two hours. |
| 2 | A person waits in a hospital corridor for test results. |
| 3 | Commuters stand on a platform as a train approaches. |
| 4 | A student reviews notes in a quiet library. |
| 5 | An airplane taxis to the runway before takeoff. |
| 6 | A cashier scans items at a grocery checkout. |
| 7 | A developer reviews logs after a test run. |
| 8 | A teacher distributes worksheets at the start of class. |
| 9 | A barista prepares orders during the morning rush. |
| 10 | A receptionist schedules appointments over the phone. |
| 11 | A family packs boxes before moving day. |
| 12 | A scientist calibrates equipment in a lab. |
| 13 | A gardener waters plants in a public park. |
| 14 | A journalist transcribes an interview recording. |
| 15 | A driver stops at a red light on a city street. |
| 16 | A warehouse team inventories newly arrived shipments. |
| 17 | An artist arranges brushes and paints before starting. |
| 18 | A person fills out forms at a government office. |
| 19 | A traveler reads the departure board at the airport. |
| 20 | A runner ties shoelaces at the starting line. |
| 21 | An engineer attends a project stand-up meeting. |
| 22 | Two customers sign paperwork at a bank desk. |
| 23 | A chef checks ingredients in a restaurant kitchen. |
| 24 | A delivery courier organizes packages inside a van. |
| 25 | A photographer sets up a tripod in a museum hall. |
| 26 | A patient sits in a clinic waiting room. |
| 27 | A voter stands in line at a polling place. |
| 28 | A swimmer steps onto the pool deck for practice. |
| 29 | A musician tunes a guitar before rehearsal. |
| 30 | A researcher opens survey responses on a laptop. |
| 31 | A family sits at a dining table during dinner. |
| 32 | A person tidies a small apartment on a weekend morning. |
| 33 | A conference audience listens to a keynote address. |
| 34 | A cyclist locks a bike outside a store. |
| 35 | A volunteer sorts canned goods at a community pantry. |
| 36 | A hiker studies a trail map at a junction. |
| 37 | An office worker files documents in a cabinet. |
| 38 | A neighbor takes out recycling bins to the curb. |
| 39 | A child builds a model with plastic blocks. |
| 40 | A librarian reshelves returned books. |
| 41 | A team reviews a quarterly report in a conference room. |
| 42 | A passenger scans a ticket at the station gate. |
| 43 | A doctor reviews a chart before entering the exam room. |
| 44 | A software team deploys an update after code review. |
| 45 | A shopper compares prices on two similar products. |
| 46 | A resident waters houseplants near a window. |
| 47 | A student submits an assignment before midnight. |
| 48 | A person renews a passport at a service counter. |
| 49 | A tourist photographs a landmark from a viewing deck. |
| 50 | A person waits for a ride-share pickup at the curb. |

## G  Speech Experiments

We generated usage-annotated text prompts and synthesized audio with CosyVoice2 (Du et al., 2024), a open-source controllable speech synthesis model Four usage dimensions were considered: prosody, topic, context, and genre. For each usage we created balanced positive and negative examples. Unless otherwise noted, the speech was rendered in a neutral voice so that the variation came from the text rather than vocal affect.

**Prosody (delivery)**   Semantic content was held constant (e.g., "the experiment passed/failed"), while affective wording varied. Positive examples included *excited, warm, calm, relieved, confident, grateful*; negative examples included *frustrated, sad, flat, worried, drained, critical*. During synthesis, CosyVoice2 was instructed to render positive utterances with higher pitch and energy, and negative utterances with a subdued, lower-pitched delivery.

**Topic (semantic domain)**   We varied the subject matter while keeping style neutral. Domains included *movie, travel, food, sports, work, music, study, health*, each with positive and negative versions (e.g., "The concert was uplifting" vs. "The sound was uninspired"). Speech was synthesized neutrally to ensure differences stemmed from semantics.

**Context (communicative setting)**   We varied the discourse situation, covering *friends' chat, formal talk, manager briefing, supportive message, all-hands announcement*. Templates reflected pragmatic conventions (e.g., "Good news:" vs. "I regret to inform you...\"). Speech was generated with a neutral delivery.

**Genre (register/style)**   We varied stylistic register, covering *tweet, news, novel, email, stand-up update, blog*. Each followed its own conventions (e.g., tweets with emojis, news in a factual tone, novels with narrative description), with both positive and negative variants. Speech was synthesized neutrally so that differences came from style.

**Additional Processing**   To avoid overly short or templated sentences, seeds were stochastically extended with temporal/place details, process descriptions, and booster clauses consistent with the sentiment polarity. Duplicates were removed, and the dataset was balanced across usage–sentiment buckets. CosyVoice2 produced 16kHz mono WAV files using male and female reference voices. Metadata included utterance ID, path, label, speaker, usage, variant, and original text.

**Summary**   In prosody, we manipulated *acoustic delivery* while keeping semantics constant. In topic, context, and genre, we manipulated *textual content or style* while keeping delivery neutral. This design isolates different usage factors in both text and speech.

**Table 15:** Summary of usage dimensions for synthetic speech.

| Usage | What varied | Examples | How realized |
|---|---|---|---|
| Prosody | Acoustic delivery | Positive: excited, warm; Negative: sad, drained | Controlled by TTS tone (pitch, energy) |
| Topic | Semantic domain | Movie, travel, food, sports, etc. | Varied in text content; neutral TTS delivery |
| Context | Communicative setting | Friends' chat, formal talk, manager briefing | Varied in text pragmatics; neutral TTS delivery |
| Genre | Register / style | Tweet, news, novel, email, blog, stand-up | Varied in text style; neutral TTS delivery |

**Evaluation**   We use the IEMOCAP corpus (Busso et al., 2008) and map categorical labels into binary sentiment. The positive class includes happy and excited, while the negative class covers angry, sad, frustrated, disgusted, and fearful. The surprised category is excluded due to ambiguous polarity.

Table 16 shows the top-10 results from our grid search on IEMOCAP. The best-performing configurations consistently peak at shallow LLM layers (e.g., layer 3). This contrasts with the text setting,

| Combination | Best layer | Acc | F1 | AUROC | ΔAcc |
|---|---|---|---|---|---|
| M+P+T (1.5:1.0:2.0) | 3 | 0.787 | 0.666 | 0.827 | +0.026 |
| M+P+T (1.0:1.0:2.0) | 3 | 0.787 | 0.662 | 0.828 | +0.026 |
| M+P+T (2.0:1.0:2.0) | 3 | 0.786 | 0.668 | 0.828 | +0.025 |
| M+P+T+G+C (1.0:1.0:0.25:0.25:2.0) | 3 | 0.786 | 0.659 | 0.826 | +0.025 |
| M+P+T+G+C (0.25:1.0:0.5:1.0:2.0) | 2 | 0.786 | 0.670 | 0.827 | +0.025 |
| M+P+T (0.5:1.0:2.0) | 3 | 0.786 | 0.653 | 0.827 | +0.025 |
| M+P+T+G+C (0.25:1.0:1.0:1.0:2.0) | 2 | 0.786 | 0.670 | 0.825 | +0.024 |
| M+P+T+G+C (1.5:1.0:0.25:0.25:2.0) | 3 | 0.786 | 0.662 | 0.827 | +0.024 |
| M+P+T+G+C (1.5:1.0:0.25:0.5:2.0) | 3 | 0.786 | 0.663 | 0.827 | +0.024 |
| M+P+T+G+C (0.25:1.0:0.25:0.5:2.0) | 2 | 0.785 | 0.678 | 0.826 | +0.024 |

**Table 16:** Top-10 results from the grid search on IEMOCAP (Session 5). Abbreviations: M=main, P=prosody, T=topic, C=context, G=genre.

where main-only and usage-only axes typically reach their optimum in mid-level layers after semantic integration. The difference reflects how sentiment cues are distributed across modalities: in audio, the encoder already extracts rich acoustic and prosodic patterns, and the first few LLM layers directly preserve this information before it becomes abstracted into higher-level semantics. Consequently, usage and polarity signals can be captured more effectively at shallow depths, whereas deeper layers gradually dilute prosodic cues as they align with semantic or generative objectives.

## USE OF LARGE LANGUAGE MODELS

We used LLMs in three limited ways: (1) to generate synthetic text data for constructing usage-annotated probing datasets, (2) to synthesize audio data for preliminary cross-modal experiments, and (3) to polish the readability of the manuscript. All research ideas, methodological designs, experiments, and analyses were carried out independently by the authors, and the use of LLMs does not affect the validity of our findings.

