# OpenReview forum: "Usage-Aware Sentiment Representations in Large Language Models"
_ICLR.cc/2026/Conference — Submitted to ICLR 2026_

### Official Review · Reviewer_3raM · 2025-10-22

**Soundness:** 1
**Presentation:** 1
**Contribution:** 2
**Rating:** 2
**Confidence:** 4

**Summary:**

In this paper, the authors investigate how sentiment is represented in Large Language Models (LLMs). They argue that a single, universal sentiment "direction" is insufficient and propose a "usage-aware" framework that incorporates linguistic factors like tone, topic, context, and genre. They train linear probes on both a pooled dataset and usage-specific datasets, showing that combining these probes leads to marginally improved sentiment classification performance. They also analyze sentiment at the neuron level, identifying "usage-invariant" and "usage-sensitive" neurons. Finally, they demonstrate that these derived axes can be used for activation steering, and that the component of a usage-axis orthogonal to the main sentiment axis can modulate stylistic elements without altering the core sentiment.

**Strengths:**

See below

**Weaknesses:**

## Overall Rating
I recommend rejecting this paper. I found the paper very difficult to follow, with key methodological details unclear. Based on my current understanding, the core results are either marginal performance gains that are not particularly surprising, or novel claims that lack rigorous evidence. I am open to the possibility that I have misunderstood key aspects of the work and would welcome clarification from the authors, but in its current state, I cannot recommend acceptance.

## Major Comments

*The following are things that, if adequately addressed, may change my score.*

1.  **Methodological Clarity:** The paper is difficult to parse due to a lack of clarity on crucial methodological points, making it harder to interpret the results.
    *   **"Usage-Specific" Datasets:** The concept of a "usage-specific" dataset is core to the paper but is not clearly defined until the appendices (line 675). The main text should precisely explain how these datasets are constructed and how they make a particular usage factor salient for a sentiment classification task. My initial assumption was that a "tone" probe would be trained on data of a single tone, or to predict tone itself. Instead, it appears a single probe is trained on a collection of texts with varied tones, with the hope that tone is the most salient differentiating factor.
    *   **"Main+Sub" Combination:** The paper's central results rely on a "Main+Sub" axis combination (e.g., Table 1, line 217), but this operation is never defined. Assuming it is vector addition, this is not a standard or principled method for combining probe directions, and its meaning is unclear. The paper should define this operation and justify its use before any conclusions can be drawn from it.

2.  **Limited Significance of Classification Results:** The main quantitative result—that "Main+Sub" axes slightly outperform the "Main" axis alone—is not particularly surprising. I do not find this result surprising. Supervised methods like linear probing generally perform better when the training data is more similar to the evaluation data. By creating more specialized probes and combining them, it is expected that they would better cover the nuances of diverse evaluation sets, leading to marginal gains.

3.  **Confounded Geometric Analysis:** The authors claim that because the "main axis lies within the usage subspace" (lines 413-417), sentiment is "largely shaped by usage-conditioned variation." This conclusion is built on a confounded experimental design. The "main" probe was trained on the union of the datasets used for the usage-specific probes. It is therefore tautological that the resulting "main" vector would be well-approximated by the span of the usage vectors. Even if it were constructed with separate datasets, this observation does not rule out the simpler hypothesis that there is a single primary sentiment direction and the usage-specific probes are merely noisy measurements of it.

4.  **Qualitative Steering Evidence is Unconvincing**: The idea that steering with the orthogonal component of a usage axis can alter style without changing sentiment (Table 5, line 388) is potentially interesting. However, the qualitative examples provided are not compelling enough to serve as strong evidence. The effect seems subtle, and it is difficult to judge its consistency and magnitude from a few examples. The authors do not state whether these examples were cherry-picked or selected randomly. To be compelling, this claim requires a rigorous, large-scale evaluation, for example using an LLM judge to rate sentiment and stylistic attributes across a randomized set of prompts, to demonstrate a statistically significant and meaningful effect.

**Questions:**

See above

---

> ### Author Response · Authors · 2025-11-21
>
> We appreciate your careful review of our work and the constructive comments provided.
>
> > Q1.1: "Usage-Specific" Datasets: The concept of a "usage-specific" dataset is core to the paper but is not clearly defined until the appendices (line 675). The main text should precisely explain how these datasets are constructed and how they make a particular usage factor salient for a sentiment classification task. My initial assumption was that a "tone" probe would be trained on data of a single tone, or to predict tone itself. Instead, it appears a single probe is trained on a collection of texts with varied tones, with the hope that tone is the most salient differentiating factor.
>
> We recognize that the main text did not clearly explain how a usage-specific dataset is constructed. In our setup, for example, the tone-specific dataset is created through controlled generation (Table 8), where each prompt explicitly enforces a particular tone usage condition (e.g., Sincere, Sarcastic, Formal). Within each tone substyle, we generate balanced positive and negative sentiment examples, Thus, the probe is trained to distinguish sentiment within each tone category. This tone conditioning produces clearer and more consistently expressed tone signals than those found in naturally mixed utterances, making tone the primary structured source of non-sentiment variation by construction. We will revise the main text to clarify this construction.
>
> > Q1.2: "Main+Sub" Combination: The paper's central results rely on a "Main+Sub" axis combination (e.g., Table 1, line 217), but this operation is never defined. Assuming it is vector addition, this is not a standard or principled method for combining probe directions, and its meaning is unclear. The paper should define this operation and justify its use before any conclusions can be drawn from it.
>
> We clarify that the operation is neither arbitrary nor unprincipled. In our method, “Main+Sub” is implemented as a mean-pooled and normalized linear combination of the main sentiment axis and the selected usage-specific axes, i.e., $v_{\text{Main+Sub}}=\frac{\frac{1}{k+1}\sum_{i=0}^{k}v_i}{\left\lVert \frac{1}{k+1}\sum_{i=0}^{k}v_i\right\rVert}$
>
> This formulation follows established practice in representation learning: prior work routinely constructs semantic directions by averaging or linearly combining multiple seed-based or probe-based vectors.
>
> For example, Bolukbasi et al. (2016) derive gender directions by aggregating multiple word-pair difference vectors and projecting them via PCA, which is a linear combination of directional components; Erk & Apidianaki (2024) construct interpretable semantic dimensions by averaging multiple seed-derived directions and then adjusting them using human judgments; and Coates & Bollegala (2018) show that simple vector averaging yields stable and robust semantic directions in embedding space. Our use of a normalized mean combination is therefore consistent with these principled approaches and provides a straightforward, interpretable way to integrate multiple semantic axes.
>
> *Joshua Coates and Danushka Bollegala. 2018. Frustratingly Easy Meta-Embedding – Computing Meta-Embeddings by Averaging Source Word Embeddings. In Proceedings of the 2018 Conference of the North American Chapter of the Association for Computational Linguistics: Human Language Technologies, Volume 2 (Short Papers), pages 194–198, New Orleans, Louisiana. Association for Computational Linguistics.*
>
> *Bolukbasi, T., Chang, K. W., Zou, J. Y., Saligrama, V., & Kalai, A. T. (2016). Man is to computer programmer as woman is to homemaker? debiasing word embeddings. Advances in neural information processing systems, 29.*
>
> *Katrin Erk and Marianna Apidianaki. 2024. [Adjusting Interpretable Dimensions in Embedding Space with Human Judgments](https://aclanthology.org/2024.naacl-long.146/). In *Proceedings of the 2024 Conference of the North American Chapter of the Association for Computational Linguistics: Human Language Technologies (Volume 1: Long Papers)*, pages 2675–2686, Mexico City, Mexico. Association for Computational Linguistics.*

---

> ### Author Response · Authors · 2025-11-21
> **Response (continued)**
>
> > Q2: Limited Significance of Classification Results: The main quantitative result—that "Main+Sub" axes slightly outperform the "Main" axis alone—is not particularly surprising. Supervised methods like linear probing generally perform better when the training data is more similar to the evaluation data. By creating more specialized probes and combining them, it is expected that they would better cover the nuances of diverse evaluation sets, leading to marginal gains.
>
> Our probes are trained entirely on synthetic, LLM-generated text and evaluated across multiple real-world datasets. Importantly, these test datasets exhibit diverse and non-uniform writing styles, so the observed gains cannot be attributed to an incidental match between the synthetic training distribution and any single test-set style. The fact that the combined main axis and usage-aware axes improve performance consistently across these varied datasets indicates that this composite representation captures sentiment variation that generalizes beyond the synthetic training distribution.
>
> Several gains are substantial in practice—for example, +11.71% (LLaMA on Cringe), +11.95% (Gemma on OkCupid), and +6.92% (Mistral on OkCupid). These results suggest that usage-aware axes reveal sentiment variation that is harder to detect when training on a fully pooled dataset.
>
> > Q3: Confounded Geometric Analysis: The authors claim that because the "main axis lies within the usage subspace" (lines 413-417), sentiment is "largely shaped by usage-conditioned variation." This conclusion is built on a confounded experimental design. The "main" probe was trained on the union of the datasets used for the usage-specific probes. It is therefore tautological that the resulting "main" vector would be well-approximated by the span of the usage vectors. Even if it were constructed with separate datasets, this observation does not rule out the simpler hypothesis that there is a single primary sentiment direction and the usage-specific probes are merely noisy measurements of it.
>
> If the usage-specific probes were merely noisy estimates of a single underlying direction, their vectors would be nearly collinear. In practice they show only moderate cosine similarity, not the near-1.0 alignment we would expect under that hypothesis, which indicates meaningful usage-conditioned variation rather than small noisy perturbations of one axis.
>
> Neuron-level evidence also challenges the idea that the usage-specific axes are merely noisy variants of a single underlying direction. We observe two clear groups of neurons: usage-sensitive neurons whose polarity flips when we compare different usage families, and usage-invariant neurons whose polarity remains stable. These polarity patterns are consistent across usage families and datasets, indicating that they are not random noise.
>
> We also ran a new targeted neuron-ablation experiment that separately removes usage-sensitive and usage-invariant neurons while measuring the LLM’s own sentiment predictions. The two groups produce markedly different effects: ablating usage-sensitive neurons yields large accuracy drops in early layers (e.g., up to 0.49), whereas ablating usage-invariant neurons produces only minor effects when removed alone. When both groups are ablated together, accuracy degrades broadly across early and mid layers (up to 0.96). These findings provide evidence that is more consistent with structured usage-conditioned variation than with a single latent direction plus noise.

---

> ### Author Response · Authors · 2025-11-21
> **Response (continued)**
>
> > Q4: Qualitative Steering Evidence is Unconvincing: The idea that steering with the orthogonal component of a usage axis can alter style without changing sentiment (Table 5, line 388) is potentially interesting. However, the qualitative examples provided are not compelling enough to serve as strong evidence. The effect seems subtle, and it is difficult to judge its consistency and magnitude from a few examples. The authors do not state whether these examples were cherry-picked or selected randomly. To be compelling, this claim requires a rigorous, large-scale evaluation, for example using an LLM judge to rate sentiment and stylistic attributes across a randomized set of prompts, to demonstrate a statistically significant and meaningful effect.
>
> We appreciate the reviewer’s comment and agree that the qualitative examples alone are insufficient to support a strong claim. To strengthen this point, we conducted a small quantitative follow-up using GPT-4o as an automatic judge. For the tone usage factor (50 prompts), steering with the raw usage vector changed tone in 84% of cases and altered sentiment in 46% of cases. Steering with the orthogonal component increased tone changes to 90% while reducing sentiment drift to 34%. Although this is a preliminary experiment, it supports the basic intuition that orthogonal usage components can adjust style while better preserving sentiment than the raw direction.
>
> | Steering Mode        | Tone Changed | Tone % | Sentiment Changed | Sentiment % |
> |----------------------|--------------|--------|--------------------|--------------|
> | Raw usage vector     | 42 / 50      | 84%    | 23 / 50            | 46%          |
> | Orthogonal component | 45 / 50      | 90%    | 17 / 50            | 34%          |

---

> > ### Comment · Reviewer_3raM · 2025-11-22
> >
> > Thanks to the authors for the detailed reply and the follow-up experiment re style steering. I unfortunately will retain my score

---

> > > ### Author Response · Authors · 2025-11-22
> > >
> > > Thank you for your comment. Do you mind explanining if our response answered your question? If not, we'd be happy to clarify further.

---

### Official Review · Reviewer_4QNq · 2025-10-27

**Soundness:** 3
**Presentation:** 2
**Contribution:** 2
**Rating:** 4
**Confidence:** 3

**Summary:**

# Summary
This paper proposes a Usage-Aware affective representation framework to address the instability of probed sentiment directions across datasets, which harms downstream reliability. The authors attribute this variability to linguistic usage factors—such as tone, topic, context, and genre—and introduce two complementary analyses:

- Axis Level: A usage-invariant axis is derived as the intersection of sentiment axes from aggregated and usage-specific data, capturing core affective signals.
- Neuron Level: Neurons are classified as usage-invariant or usage-sensitive for fine-grained interpretation.

**Strengths:**

# Strengths
1. The paper explicitly identifies the reliability issue of probe-derived sentiment axes in cross-dataset applications and innovatively links this problem to linguistic "usage" factors, offering a theoretically grounded explanation for variability in representation learning.

2. The framework conducts complementary analyses at both the axis and neuron levels. The intersection operation at the axis level is the core contribution addressing cross-dataset consistency, while the distinction at the neuron level provides a fine-grained understanding of the model’s internal mechanisms.

**Weaknesses:**

# Weaknesses
1. This is the biggest barrier to the practical deployment of the framework. The core of the method—computing the intersection between usage-specific axes and the invariant axis—relies heavily on fine-grained, high-quality labels of usage factors (e.g., tone, genre, topic) in the training data. In real-world scenarios, acquiring such detailed, multi-dimensional annotations is extremely costly, difficult to label consistently, and prone to low inter-annotator agreement. This severely limits the method’s potential for real-world adoption and scalability.

2. The paper introduces four usage dimensions—genre, tone, context, and topic—but does not sufficiently justify their mutual exclusivity or independence. For example, tone and genre may be highly correlated (e.g., tweets often inherently carry an informal tone). Without theoretical or empirical evidence demonstrating that these dimensions can be treated as orthogonal, decomposing sentiment representations along them risks conflating interdependent factors, potentially inflating the perceived benefit of usage-specific axes.

3. The authors should analyze the distribution of usage factors across the benchmark datasets. It is possible that the observed performance gains stem not from the proposed method itself, but from alignment between the usage distributions in the training data and those in the evaluation benchmarks.

4. Although probes are trained at every layer, the paper only reports aggregated results and does not investigate which layers benefit most from usage-specific axes. Understanding how usage-aware sentiment signals evolve across model depth would significantly strengthen the interpretability of the representations and provide practical guidance for deployment (e.g., which layer to use for feature extraction).

5. typos: page 16, line 831, "We use instruction-tuned versions of three popular decoder-only LLMs." repeat.

**Questions:**

Refer to Weakness.

---

> ### Author Response · Authors · 2025-11-21
>
> We appreciate the reviewer for the thoughtful feedback.
>
> > W1: This is the biggest barrier to the practical deployment of the framework. The core of the method—computing the intersection between usage-specific axes and the invariant axis—relies heavily on fine-grained, high-quality labels of usage factors (e.g., tone, genre, topic) in the training data. In real-world scenarios, acquiring such detailed, multi-dimensional annotations is extremely costly, difficult to label consistently, and prone to low inter-annotator agreement. This severely limits the method’s potential for real-world adoption and scalability.
>
> All usage labels (tone, topic, genre, context) are automatically generated together with the synthetic training data by GPT-4o, so no costly or inconsistent manual annotation is required. Moreover, usage labels are only needed during training to learn usage-specific axes; the method does not require any usage annotations at test time, and can be directly applied to unseen real-world inputs.
>
> > W2: The paper introduces four usage dimensions—genre, tone, context, and topic—but does not sufficiently justify their mutual exclusivity or independence. For example, tone and genre may be highly correlated (e.g., tweets often inherently carry an informal tone). Without theoretical or empirical evidence demonstrating that these dimensions can be treated as orthogonal, decomposing sentiment representations along them risks conflating interdependent factors, potentially inflating the perceived benefit of usage-specific axes.
>
>
> Our method does not assume that the four usage factors are mutually exclusive, independent, or orthogonal. In natural language, usage dimensions such as tone, genre, topic, and context naturally co-occur and correlate. We use usage factors only to construct more interpretable sentiment axes, not to recover independent latent variables, so independence is neither required nor conceptually expected.
>
> > W3: The authors should analyze the distribution of usage factors across the benchmark datasets. It is possible that the observed performance gains stem not from the proposed method itself, but from alignment between the usage distributions in the training data and those in the evaluation benchmarks.
>
>  Test datasets are real-world datasets whose usage-factor distributions are not consistent with one another. As shown in Figure 6, the tone, topic, genre, and context factors vary substantially across the evaluation datasets. This diversity indicates that the observed gains cannot be explained by an accidental match in usage composition.
>
> > W4: Although probes are trained at every layer, the paper only reports aggregated results and does not investigate which layers benefit most from usage-specific axes. Understanding how usage-aware sentiment signals evolve across model depth would significantly strengthen the interpretability of the representations and provide practical guidance for deployment (e.g., which layer to use for feature extraction).
>
> We agree that understanding the layerwise evolution of usage-aware sentiment signals is important for interpretability. To address this point, we conducted a per-layer analysis across datasets and usage combinations. While the exact gains vary, a consistent descriptive pattern emerges: sentiment probing accuracy, tends to peak in the mid layers
>
> > W5: typos: page 16, line 831, "We use instruction-tuned versions of three popular decoder-only LLMs." repeat.
>
> Thank you for pointing this out—we have fixed the repetition.

---

### Official Review · Reviewer_AZkL · 2025-10-28

**Soundness:** 2
**Presentation:** 2
**Contribution:** 2
**Rating:** 6
**Confidence:** 2

**Summary:**

This paper introduces a linguistically grounded framework for modeling sentiment representations in LLMs through usage-aware axes. By decomposing sentiment into usage-specific factors and analyzing both axis- and neuron-level behaviors, the study enhances interpretability and reliability.

**Strengths:**

1. The core idea grounds sentiment variability in explicit linguistic usage factors, offering a highly interpretable framework for LLM sentiment analysis, superior to distributional methods that sacrifice linguistic meaning.

2. The usage-aware axes achieved a substantial average improvement in cross-domain sentiment classification accuracy, demonstrating superior robustness and transferability.

**Weaknesses:**

1. The core dataset is synthetically generated by ChatGPT-4.0, which risks introducing model-induced biases and may fail to capture the subtle linguistic nuances of natural human-generated text.

2. The analysis is restricted to four predefined usage factors (tone, topic, context, genre), neglecting other key influences on sentiment variability, such as sarcasm or cultural context.

**Questions:**

Please see the Weaknesses.

---

> ### Author Response · Authors · 2025-11-21
>
> Thank you for the constructive review and for the helpful feedback. We address the raised concerns below.
>
> > W1: The core dataset is synthetically generated by ChatGPT-4.0, which risks introducing model-induced biases and may fail to capture the subtle linguistic nuances of natural human-generated text.
> >
>
> We refer to Li et al. (EMNLP 2023), which systematically evaluates LLM-generated synthetic data versus human-annotated data across multiple text classification tasks. They show that when a task has low subjectivity, such as sentiment polarity with clear lexical cues, models trained solely on zero-shot synthetic data perform comparably to those trained on human-labeled data. Conversely, performance diverges only for highly subjective tasks. Since our task involves binary sentiment classification with explicit sentiment markers, it falls into the low-subjectivity regime identified by Li et al. Therefore, using synthetic data generated by a stronger model (GPT-4o) is reasonable and expected to yield high-fidelity training examples for this setting.
>
> In addition, our experiments evaluate the synthetic data not on a single downstream model, but across three architecturally distinct LLMs. If the GPT-4o–generated data contained strong model-induced biases, we would expect different model architectures to exhibit inconsistent behavior when trained or evaluated on it. In contrast, all three models show the same qualitative patterns, indicating that the synthetic data captures generalizable sentiment structure rather than artifacts tied to the model that generated it.
>
> *Zhuoyan Li, Hangxiao Zhu, Zhuoran Lu, and Ming Yin. 2023. [Synthetic Data Generation with Large Language Models for Text Classification: Potential and Limitations](https://aclanthology.org/2023.emnlp-main.647/). In Proceedings of the 2023 Conference on Empirical Methods in Natural Language Processing, pages 10443–10461, Singapore. Association for Computational Linguistics.*
>
> > W2: The analysis is restricted to four predefined usage factors (tone, topic, context, genre), neglecting other key influences on sentiment variability, such as sarcasm or cultural context.
>
> Sentiment variability can be influenced by many additional factors such as sarcasm, cultural background, or pragmatic intent. However, our goal is not to enumerate all possible sources of variation, but to ground the analysis in a set of core and widely recognized linguistic usage factors. The four dimensions we use—tone, topic, context, and genre—are directly motivated by prior linguistic and sentiment research (e.g., Ousidhoum et al., 2019; Blitzer et al., 2007; Joshi et al., 2015; Barnes et al., 2017), and have been repeatedly shown to account for major and cross-domain sources of sentiment variation in human communication.
>
> Higher-level phenomena such as sarcasm or culturally shaped sentiment expression are certainly important, but they are usually understood as combinations or refinements of more basic usage factors like tone and context rather than as separate universal sentiment dimensions. Our framework can naturally incorporate such additional factors, but in this work we focus on four core usage dimensions as a principled and interpretable starting point rather than a complete taxonomy.

---

> ### Comment · Reviewer_AZkL · 2025-11-22
>
> Thank you for the extended explanations. I decide to remain my score.

---

### Official Review · Reviewer_125c · 2025-10-30

**Soundness:** 3
**Presentation:** 3
**Contribution:** 2
**Rating:** 4
**Confidence:** 3

**Summary:**

This paper proposed a usage-aware sentiment representation framework for LLMs that grounds sentiment variability in linguistic usage factors such as tone, topic, context, and genre.

The approach was motivated by two limitations of existing methods: the instability of sentiment directions extracted via linear probes, and the lack of interpretability in distributional representations such as Gaussian subspaces.

The authors introduced a two-level analysis: (1) at the axis level, constructing both pooled and usage-specific sentiment directions to examine how linguistic usage influences representational reliability; and (2) at the neuron level, differentiating usage-invariant from usage-sensitive neurons to reveal finer-grained encoding patterns.

Empirical results showed that usage-aware sentiment representations improve both classification accuracy and controllability of sentiment steering.

**Strengths:**

The work addresses a feasible limitation of current sentiment representation extraction methods: linear probes lack reliability, while more complex methods lack interpretability.

The authors showed that the natural variability and interpretability of sentiment lie in linguistic usage factors and designed a two-level framework to capture higher-quality sentiment representations, through sentiment-guided axis construction and neuron-level analysis.

The writing is clear and logically structured, and the experimental design is coherent, complete, and solid. The application of fine-grained representation analysis to the sentiment domain appears original and contributes meaningfully to decoding factors that contribute to sentiment-based interpretability in LLMs.

**Weaknesses:**

It is not particularly surprising for me that extracting separate linear probes for different usage-specific axes yields more specialized representations that perform better at classification and steering. This essentially demonstrates that representations separated by linguistic patterns possess greater predictive power for sentiment.

This raises some concerns about conceptual novelty. While the framework provides a linguistically motivated decomposition of sentiment axes, it mainly repackages an established idea—linear disentanglement by conditioning on auxiliary attributes—into a sentiment-specific context. Prior studies on domain adaptation and representation disentanglement (e.g., Blitzer et al., 2007 (https://aclanthology.org/P07-1056.pdf); Hewitt & Manning, 2019 (https://aclanthology.org/N19-1419.pdf)) have already shown that task-specific subspaces can improve generalization. To strengthen the contribution, the authors may provide a clearer theoretical justification of what “usage-awareness” adds beyond such conditional probing and what do the results imply.

**Questions:**

1. What do your results imply about the role of linguistic usage factors in sentiment interpretability? Do the axis- and neuron-level analyses jointly suggest that usage factors causally structure sentiment representations in LLMs, or mainly correlate with them? Adding such discussion could be beneficial.

2. Can you provide causal evidence that “usage-invariant” and “usage-sensitive” neurons drive predictions rather than merely correlate with them? Further analyses using targeted neuron ablations, activation patching, or causal mediation methods could strengthen the causal claims.

---

> ### Author Response · Authors · 2025-11-21
>
> Thank you for the careful reading of our paper and for providing constructive and insightful reviews.
> > W1: It is not particularly surprising that extracting separate linear probes for different usage-specific axes yields more specialized representations that perform better at classification and steering. This essentially demonstrates that representations separated by linguistic patterns possess greater predictive power for sentiment.
>
> We appreciate the reviewer’s perspective. Our goal is not to claim that usage-specific probes are surprising in the sense of yielding higher accuracy. Rather, our main objective is to build interpretable sentiment axes that are grounded in linguistically defined usage factors.
>
> By deriving axes from subsets organized by tone, topic, genre, and context, each sentiment direction is anchored in a usage subset, rather than an undifferentiated mixture of examples. This prevents heterogeneous patterns from being averaged together and contributes to more reliable downstream behavior.
>
> Our experiments support this: combining the main axis with usage-aware axes yields a more reliable sentiment representation than relying on a single axis alone. In addition, our neuron-level analysis reveals distinct usage-invariant and usage-sensitive components, providing converging evidence for structured variation in the model’s internal sentiment-related representations.
>
> > W2: This raises some concerns about conceptual novelty. To strengthen the contribution, the authors may provide a clearer theoretical justification of what “usage-awareness” adds beyond such conditional probing and what do the results imply.
>
> Prior work has shown that sentiment in LLMs does not always correspond to a single fixed linear direction. Our central finding is that this variation is not arbitrary: it aligns consistently with linguistically grounded usage factors. Different usage conditions are associated with different components of the model’s internal sentiment representation.
>
> For instance, some sentiment-related neurons behave consistently across usages, while others flip polarity depending on usage. A single main axis trained on mixed data averages over these heterogeneous patterns, causing it to lose structure and become less reliable. Usage-aware axes (main + sub) preserve the usage-specific variation that is washed out in a main axis, and our flip/stable neuron analyses support this interpretation. Unlike conditional probing, we do not use usage labels at test time; usage is employed only to construct interpretable sentiment axes.
>
> >Q1: What do your results imply about the role of linguistic usage factors in sentiment interpretability? Do the axis- and neuron-level analyses jointly suggest that usage factors causally structure sentiment representations in LLMs, or mainly correlate with them? Adding such discussion could be beneficial.
>
> Our axis- and neuron-level analyses indicate that usage factors have a consistent and meaningful influence on how sentiment manifests in the model’s internal representations, rather than functioning as incidental correlations. In particular, the presence of “flipped neurons”—whose polarity shifts across usage conditions—shows that sentiment-related neural activity is usage-conditioned rather than fixed. Although we do not claim a full causal mechanism, the consistency of these usage-dependent patterns suggests that usage-aware axes capture structure that a single pooled direction loses.

---

> ### Author Response · Authors · 2025-11-21
> **Response (continued)**
>
> > Q2: Can you provide causal evidence that “usage-invariant” and “usage-sensitive” neurons drive predictions rather than merely correlate with them? Further analyses using targeted neuron ablations, activation patching, or causal mediation methods could strengthen the causal claims.
>
> Thank you for this helpful suggestion, which revealed a new insight. In response, we conducted targeted neuron ablation experiments using LLaMA-3-8B.
>
> For each example, we provided the model with a fixed sentiment-classification prompt followed by the utterance to be labeled. We then selectively ablated usage-sensitive (Flip) and usage-invariant (Stable) neurons in the model’s hidden activations at each layer during the forward pass. After the ablation, the model generated its own sentiment prediction, and we measured the accuracy drops relative to the unablated model on the DailyDialog and Confession datasets.
>
> The ablation patterns show that the two groups contribute different but complementary components of the sentiment signal:
>
> - **Flip dimensions carry strong usage-conditioned polarity cues.**
>
>   Removing them produces sharp accuracy drops in the earliest layers, indicating that the model relies heavily on these high-contrast cues during the initial stages of sentiment computation.
>
> - **Stable dimensions encode a usage-invariant component of the sentiment representation that is highly decodable.**
>
>   Stable-only probes achieve strong performance, showing that these neurons store a clean sentiment signal. However, ablating Stable neurons alone produces only small accuracy drops, meaning this stored information is not the model’s dominant inference path. The substantially larger degradation when both Stable and Flip neurons are removed indicates that Stable features provide supplementary sentiment evidence that becomes important only when primary (Flip) cues are absent.
>
> - **Ablating both groups leads to substantially greater degradation.**
>
>   Joint ablation causes accuracy to collapse across early and mid layers, demonstrating that usage-invariant (Stable) and usage-sensitive (Flip) components contribute complementary parts of the overall sentiment signal. The same pattern holds in the Confession dataset.
>
> | Ablation Type                  | Representative Layer | Accuracy Drop |
> |-------------------------------|-----------------------|----------------|
> | **Flip neurons** (usage-sensitive) | L3                    | 0.49           |
> | **Stable neurons** (usage-invariant) | L11                   | 0.08           |
> | **Flip ∪ Stable** (union)         | L3                    | **0.96**           |

---

### Meta-Review · Area_Chair_MBVp · 2026-01-07

**Summary:**

This paper argues that sentiment in LLMs is not captured by a single stable linear direction, but instead varies systematically with linguistic usage factors. The paper is clearly motivated and also empirically thorough. The neuron-level analysis and follow-up ablation experiments offer insights and make the argument more convincing. Several reviewers find the core performance gains incremental and not surprising, and question whether the framework goes beyond conditional probing already explored in prior work. Overall, while the paper presents a coherent framework with interesting neuron-level evidence, the contribution is viewed as incremental by multiple reviewers. Therefore, I recommend the rejection decision.

**Reviewer Concerns:**

Concerns from Reviewer 125c should be largely addressed by the added ablation experiments.

However, concerns from Reviewer AZkL and Reviewer 4QNq regarding novelty (and other issues) were only partially mitigated and largely remain.

**Reviewer Scores:**

Most will probably retain their original scores, some already explicitly mentioned so.

---

### Decision · Program_Chairs · 2026-01-26

Reject